# Mutant FUS causes DNA ligation defects to inhibit oxidative damage repair in Amyotrophic Lateral Sclerosis

Haibo Wang [1], Wenting Guo[2,3], Joy Mitra[1], Pavana M. Hegde [1], Tijs Vandoorne [2,3], Bradley J. Eckelmann[1,4], Sankar Mitra [1,6], Alan E. Tomkinson[5], Ludo Van Den Bosch[2,3] & Muralidhar L. Hegde [1,6,7]

Genome damage and defective repair are etiologically linked to neurodegeneration. However, the specific mechanisms involved remain enigmatic. Here, we identify defects in DNA nick ligation and oxidative damage repair in a subset of amyotrophic lateral sclerosis (ALS) patients. These defects are caused by mutations in the RNA/DNA-binding protein FUS. In healthy neurons, FUS protects the genome by facilitating PARP1-dependent recruitment of XRCC1/DNA Ligase IIIα (LigIII) to oxidized genome sites and activating LigIII via direct interaction. We discover that loss of nuclear FUS caused DNA nick ligation defects in motor neurons due to reduced recruitment of XRCC1/LigIII to DNA strand breaks. Moreover, DNA ligation defects in ALS patient-derived iPSC lines carrying FUS mutations and in motor neurons generated therefrom are rescued by CRISPR/Cas9-mediated correction of mutation. Our findings uncovered a pathway of defective DNA ligation in FUS-linked ALS and suggest that LigIII-targeted therapies may prevent or slow down disease progression.

[1] Department of Radiation Oncology, Houston Methodist Research Institute, Houston, 77030 TX, USA. [2] KU Leuven-Department of Neurosciences, Experimental Neurology and Leuven Brain Institute (LBI), Leuven, 3000, Belgium. [3] VIB, Center for Brain & Disease Research, Laboratory of Neurobiology, Leuven, 3000, Belgium. [4] Texas A&M Health Science Center, College of Medicine, Bryan, 77807 TX, USA. [5] Departments of Internal Medicine, and Molecular Genetics and Microbiology and University of New Mexico Comprehensive Cancer Center, University of New Mexico, Albuquerque, 87131 NM, USA. [6] Weill Medical College, New York, 10065 NY, USA. [7] Houston Methodist Neurological Institute, Institute of Academic Medicine, Houston Methodist, Houston, 77030 TX, USA. Correspondence and requests for materials should be addressed to M.L.H. (email: mlhegde@houstonmethodist.org)

Amyotrophic lateral sclerosis (ALS) is a neurodegenerative disease characterized by the selective and progressive death of upper and lower motor neurons. This leads to progressive muscle weakness and death of the patients usually occurs within two to five years after the onset of symptoms. In 10% of patients, there is a clear family history. The most prevalent genetic causes of familial ALS are mutations in the *Superoxide Dismutase 1 (SOD1)*, *TAR DNA Binding Protein-43 (TARDBP)*, *Fused in Sarcoma (FUS)* genes, and *Chromosome 9 Open Reading Frame 72 (C9ORF72)*. Mutations in the gene encoding the RNA/DNA-binding protein Fused in Sarcoma (FUS) have been detected in ~5% of familial ALS patients[1, 2]. These mutations are also found in a small subset (~1%) of sporadic ALS cases[3–5]. Most missense point mutations in FUS are clustered in the gene segment encoding the nuclear localization sequence (NLS) in the C-terminus and induce nuclear depletion and cytosolic aggregation of FUS[6, 7]. While arginine at position 521 (R mutated to G, H, or C) is most commonly mutated[8], the P525L mutation is associated with aggressive juvenile-onset ALS[9,10].

FUS is a multifunctional heterogeneous nuclear ribonucleoprotein (hnRNP) of the TET (TAF15, EWS, and TLS) family of RNA-binding proteins and it has been implicated in multiple aspects of RNA metabolism[7]. It is unclear yet which of these functions of FUS is critical for neurodegeneration. In healthy neurons, FUS is predominantly localized in the nucleus, but it can shuttle between the nucleus and cytosol in response to various stimuli[11,12]. FUS also binds DNA and has been recently implicated in the maintenance of genome integrity, in particular the DNA damage response (DDR) signaling, induced by DNA double-strand breaks (DSBs). In response to DSB-inducing agents, FUS is phosphorylated by ataxia-telangiectasia mutated (ATM) and DNA-dependent protein kinase (DNA-PK), which are activated by DSBs[13,14]. FUS interacts with histone deacetylase 1 (HDAC1) in primary mouse cortical neurons, which may indirectly modulate the repair of DSBs by homologous recombination (HR) and non-homologous end joining (NHEJ). Furthermore, loss of FUS abrogated both HR and NHEJ efficiency in exogenomic vector-based assays[15]. In addition, impairment of poly(ADP-ribose) polymerase (PARP)-dependent DDR signaling due to mutations in the FUS NLS, which induces cytoplasmic FUS accumulation has been linked to ALS-related neurodegeneration[16]. However, it is yet unclear which function(s) of FUS is critical for preventing neurodegeneration.

Although FUS is associated with multiple genome repair pathways, its role in the DDR is not completely understood. Independent studies demonstrated that FUS is recruited to DNA damage tracks by microirradiation (MIR) at wavelengths of 405 nm or 351 nm (UVA), respectively, in a PARP1-dependent manner presumably via interaction with PAR groups[17–19]. However, the role of FUS in downstream repair reactions was not investigated. While MIR induces clusters of different types of DNA damage, including oxidized base lesions, single-strand breaks (SSBs), and DSBs, it is generally believed that UVA damage predominantly induces SSBs via elevated reactive oxygen species (ROS)[20]. Thus, recruitment of FUS at UVA laser tracks suggests its potential role in the repair of oxidative DNA damage, which has not been thoroughly investigated.

ROS, generated endogenously as a byproduct of cellular respiration, pose a critical challenge to the genome, especially in neurons due to their high metabolic/transcriptional activity and long lifespan[21–23]. Accumulation of ROS-induced oxidized DNA bases and SSBs in affected regions of the central nervous system is associated with degenerating neurons in ALS and other neurodegenerative diseases[24,25]. These lesions are repaired by evolutionarily conserved base excision (BER) and single-strand break repair (SSBR) pathways, in which the XRCC1/LigIII complex plays a critical role in sealing DNA nicks in the final repair step[26–30].

In this study, we investigated the mechanism(s) responsible for the accumulation of SSBs in the neuronal genome after the loss of FUS. We have characterized the interaction of FUS with XRCC1/LigIII and documented that FUS facilitates the PARP1 activity-dependent recruitment of XRCC1/LigIII to oxidative DNA damage sites. The connection between FUS function and DNA ligation defects was examined in multiple model systems, including CRISPR/Cas9-mediated FUS knockout (KO) cells, familial ALS patient-derived induced pluripotent stem cells (iPSCs) with FUS mutations, motor neurons differentiated from these patient-derived iPSCs, and spinal cord tissue with FUS pathology from ALS patients. Notably, both P525L and R521H mutations in FUS cause defects in DNA ligation, albeit via distinct mechanisms. The ligation defects in FUS KO cells and patient-derived iPSC lines were rescued by addition of wild-type (WT) FUS (but not mutant FUS) and by correcting the mutation by CRISPR/Cas9, separately. Together, our results provide important molecular insights into a previously unknown DNA ligation defect in FUS-associated ALS.

## Results

**Loss of FUS induces SSB accumulation and ROS sensitization.** Although FUS is implicated in DDR, the effect of FUS deficiency on the repair of endogenous damage is unknown. To address this question, we first quantified the level of DNA SSBs vs DSBs by single cell electrophoresis (comet assay) in unstressed, FUS knockdown (KD) cells. Two independent FUS shRNAs were transfected into the neuroblastoma SH-SY5Y cell line and each shRNA induced ~70% depletion of FUS (Fig. 1a, b). The alkaline comet assay measures alkali-labile lesions, SSBs and DSBs, while the neutral comet analysis exclusively measures DSBs. Thus, a comparison of tail moments in alkaline and neutral comet assays gives a relative measure of alkali-labile lesions and SSBs vs DSBs. FUS KD cells showed an ~sixfold increase in the alkaline tail moment, but a lower than 1.5-fold change in the neutral tail moment, compared to cells transfected with control shRNA (Fig. 1c). This result implies that most unrepaired DNA strand breaks that accumulated after FUS KD were alkali-labile lesions or SSBs. The small increase in DSB level could result secondarily from closely spaced alkali-labile lesions and/or SSBs[31].

Because SSBs are induced by ROS either directly or as BER intermediates of oxidized DNA bases[32], we next tested the impact of decreased FUS expression on the cellular responses to oxidative stress. FUS KD SH-SY5Y cells were treated for 1 h with different concentrations of glucose oxidase (GO), which generates $H_2O_2$ in cellulo, mimicking endogenous ROS stress. MTT-based analysis of cellular metabolic activity, 24 h post-GO treatment showed significantly lower cell viability after FUS depletion (Fig. 1d). Similar results were obtained when the parental SH-SY5Y cells and the FUS KD derivatives were treated directly with $H_2O_2$, confirming the role of FUS in resistance to oxidative stress (Fig. 1e). In addition, clonogenic survival of oxidatively stressed HEK293 cells was significantly reduced in the absence of FUS, indicating that the protective effect of FUS against oxidative stress is not restricted to neuronal cells (Fig. 1f). We also examined the steady state level of cell viability and proliferation using the MTT and clonogenic data in unstressed control cells (Supplementary Figs. 1a, 1b). The MTT assays performed 24, 48, and 72 h after shRNA transfection revealed no significant change in cell survival, while the clonogenic assays represented as plating efficiency showed a moderate ~5% decrease in the average number of colonies formed. These data thus show that FUS KD

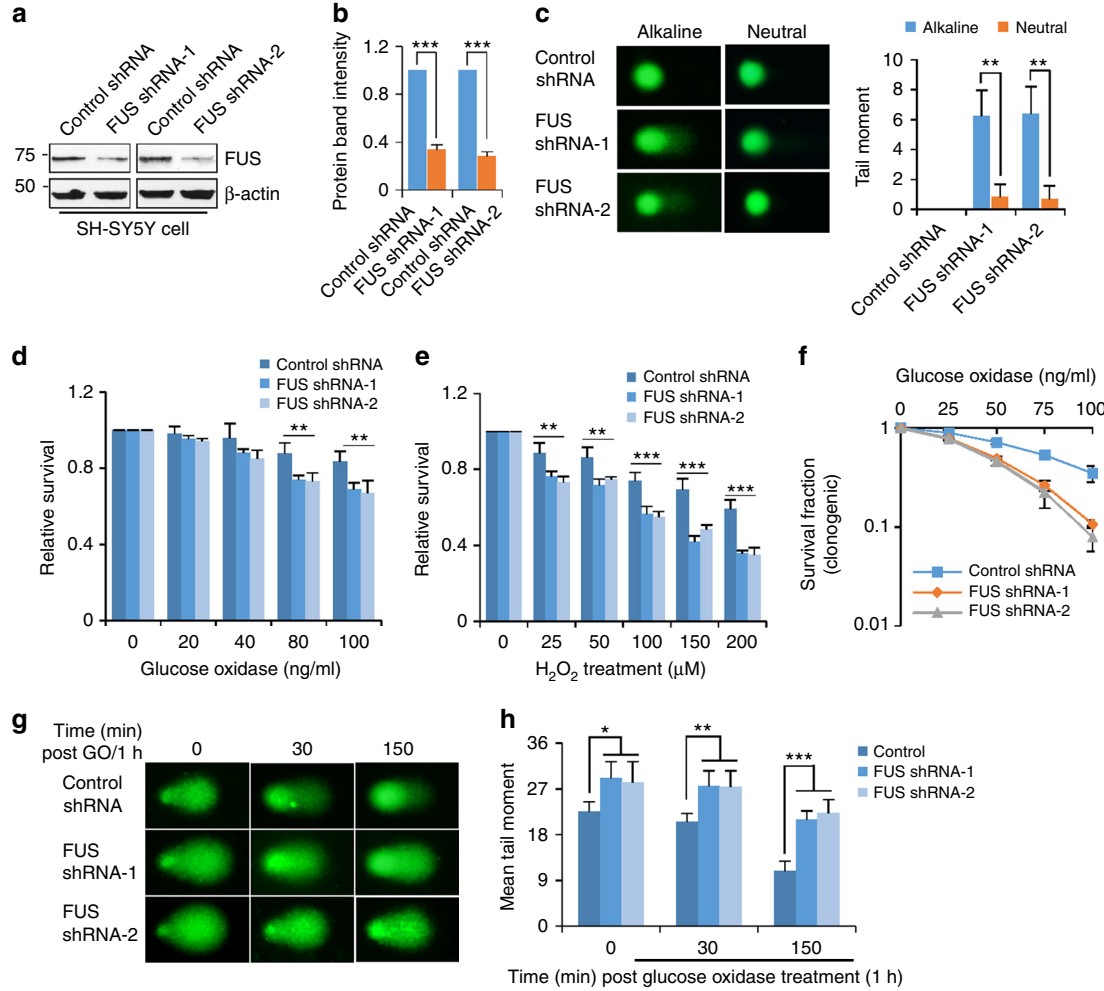

**Fig. 1** Loss of FUS induces SSB accumulation and ROS sensitization. **a**, **b** Immunoblot (IB) showing FUS knockdown (KD) by shRNAs. Total lysate were extracted from SH-SY5Y cells 48 h after the transfection with two individual FUS shRNAs. β-actin was probed as loading control. Quantitation of relative band intensity of FUS shRNA vs Control shRNA shown in **b**. The error bars are standard deviation of experiments performed in triplicate (***$p < 0.001$, two-tailed unpaired Student's $t$-test). **c** Alkaline and neutral comet assay of control vs FUS KD SH-SY5Y cells. The quantitation of mean tail moment from 50 randomly selected nuclei is shown in the histogram. The error bars are standard deviation. (**$p < 0.01$, two-tailed unpaired Student's $t$-test). **d**, **e** MTT-based viability analysis of FUS KD SH-SY5Y cells, treated with increasing doses of GO or $H_2O_2$. The cells were incubated with GO for 1 h or $H_2O_2$ as indicated dose for 3 h, 48 h after the FUS shRNA transfection. MTT assay was performed 24 h after the treatment. The error bars are standard deviation of experiment performed in triplicate (**$p < 0.01$; ***$p < 0.001$, two-tailed unpaired Student's $t$-test). **f** Clonogenic survival analysis of FUS KD HEK293 cells after GO (100 ng/ml) treatment, as in Fig. 1c. The error bars are standard deviation of experiment performed in triplicate. **g**, **h** Repair kinetics of induced oxidative genome damage after FUS KD. Alkaline comet assay of control or FUS knockdown SH-SY5Y cells at 0, 30, and 150 min post GO (100 ng/ml) treatment. Histogram represent quantitation of mean tail moment from 50 randomly selected nuclei. The error bars are standard deviation (*$p < 0.05$; **$p < 0.01$; ***$p < 0.001$, two-tailed unpaired Student's $t$-test)

did not significantly affect the survival of unstressed cells, but it only moderately affected cell proliferation.

The presence of unrepaired DNA SSBs in neuronal genomes has been linked to neurodegeneration[31–34] and is attributed to defective repair involving diverse mechanisms. To test whether the ROS sensitivity of FUS KD cells was due to impaired oxidative DNA damage repair, we evaluated the repair kinetics of alkali-labile lesions and SSBs induced by GO using alkaline comet assay. Repair was found to be significantly delayed in the FUS KD cells (Fig. 1g, h), indicating that oxidative DNA damage repair is impaired due to FUS deficiency.

**FUS forms a complex with PARP1, XRCC1, and LigIII.** To explore the involvement of FUS in repairing oxidative DNA damage, we first performed mass spectrometry analysis of

oxidative stress-dependent interaction partners of FUS isolated by co-immunoprecipitation (IP) from GO-treated SH-SY5Y cells with a FUS antibody, and detected the presence of PARP1 and XRCC1 (see Supplementary Data 1 for the list of proteins). Subsequently, analysis of Flag IPs from HEK293 cells expressing Flag-FUS revealed increased association of FUS with XRCC1, LigIII, and PARP-1, but not with other BER proteins (APE1, TDP1, PNKP, or FEN-1) in GO-treated cells (Fig. 2a). Notably, no association was observed with the DNA ligase complex, XRCC4/LigIV (Fig. 2a), involved in the repair of DSBs by NHEJ repair, or LigI (Supplementary Fig. 4c), involved in DNA replication and replication-associated excision repair, underscoring the specificity of association of FUS with XRCC1/LigIII. To investigate whether these associations also occur with endogenous FUS in neuronal cells, similar experiments were carried out with a FUS antibody and extracts from differentiated SH-SY5Y cells

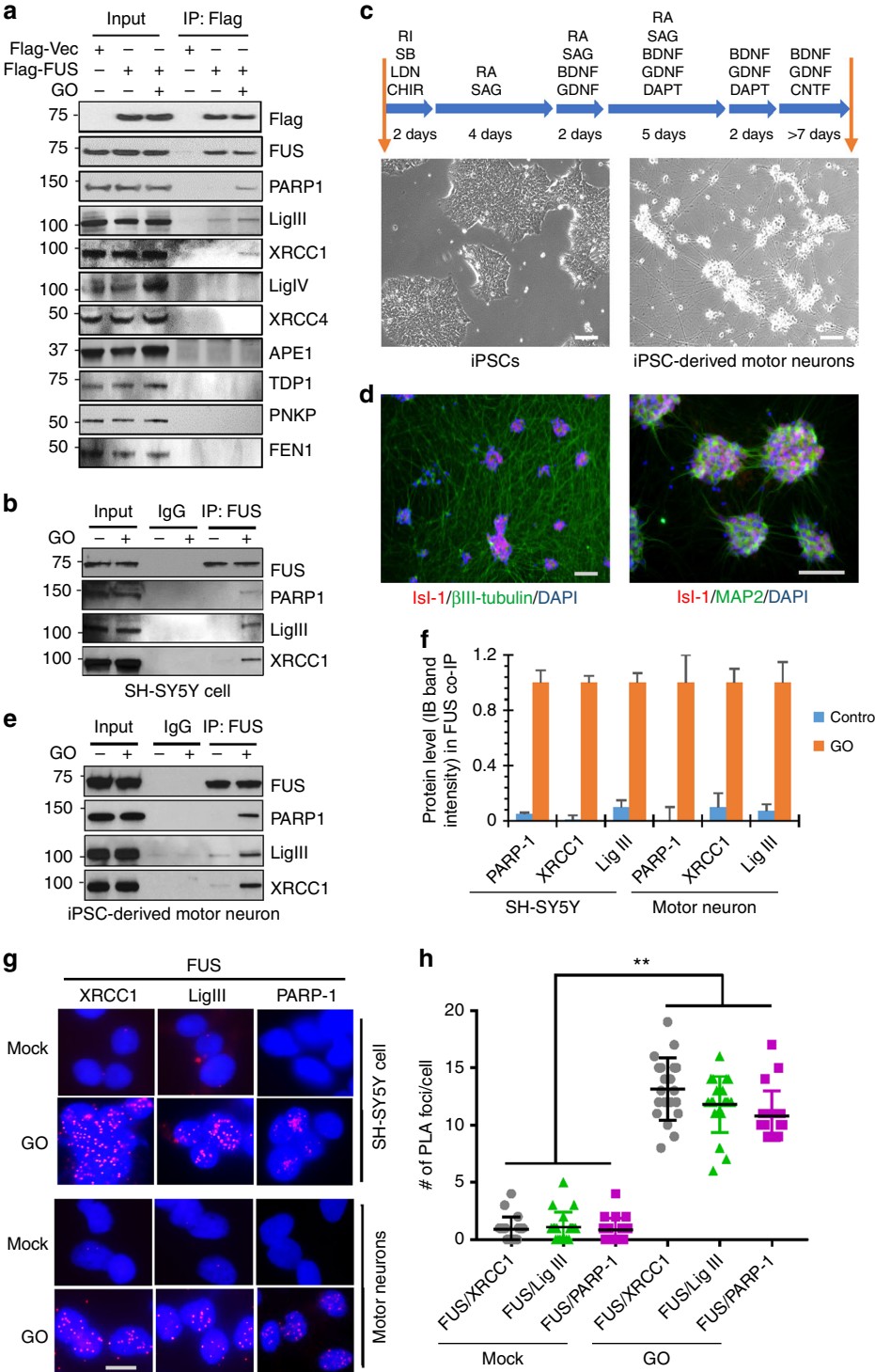

**Fig. 2** FUS forms a complex with PARP1, XRCC1, and LigIII. **a** IB of Flag-FUS co-Immunoprecipitation (co-IP) from HEK293 cells for oxidative DNA damage repair proteins with or without GO treatment (100 ng/ml for 1 h). The IP was performed with anti-Flag antibody. **b** IB of endogenous FUS co-IP from SH-SY5Y cells for PARP1, XRCC1, and LigIII. The IP was performed with anti-FUS antibody. **c** Scheme of human iPSC line differentiation to motor neurons. Representative phase-contrast images of iPSC and differentiated motor neurons are shown. Scale bar = 50 μm. **d** Immunofluorescence (IF) staining of motor neuron markers. Representative images of motor neurons that stained Isl-1, MAP2, or βIII-tubulin indicated ~80% differentiation efficiency. Scale bar = 50 μm. **e** IB of endogenous FUS co-IP from differentiated motor neurons for PARP1, XRCC1, and LigIII. **f** Quantitation of IB band intensity in Fig. 2b, e to show level change of PARP-1, XRCC1, and LigIII in FUS IP after GO treatment. The error bars are standard deviation of experiments performed in triplicate. **g** PLA of FUS vs PARP1, XRCC1, or LigIII in SH-SY5Y and iPSC-derived motor neurons with or without GO treatment. Nuclei stained with DAPI. Scale bar = 5 μm. **h** Quantitation of PLA foci from 25 motor neuronal cells (**p < 0.01, two-tailed unpaired Student's t-test). The error bars are standard deviation

(Fig. 2b) and motor neurons (Fig. 2e), which were differentiated from human iPSCs (KYOU-DXR0109B line from ATCC) using previously reported protocol[35] (Fig. 2c) that achieved up to ~80% efficiency of motor neuron differentiation detected with the neuronal marker MAP2 and specific motor neuron markers: Isl-1 and ChAT (Fig. 2d; Supplementary Fig. 2a). Under these conditions, we found that endogenous FUS associated with PARP-1 in addition LigIII and XRCC1, and that these associations were markedly enhanced in GO-treated cells (Fig. 2b–e, f). As hypothesized, FUS was also detected in reciprocal co-IPs performed with the XRCC1 antibody (Supplementary Fig. 2b). The oxidative stress-induced co-localization of FUS with PARP1, XRCC1, and LigIII was confirmed by proximity ligation assays (PLA) with differentiated SH-SY5Y cells (Fig. 2g, upper panels) and iPSC-derived motor neurons (Fig. 2g, lower panels and Fig. 2h). To confirm the specificity of the antibodies used, we performed control PLA experiments of FUS vs XRCC1 and FUS vs LigIII in control and XRCC1 or LigIII siRNA transfected cells (Supplementary Figs. 2c, 2d and 2e). As expected, there was decreased co-localization after either XRCC1 or LigIII KD, confirming the specificity of the observed interactions. Furthermore, these results are consistent with previous studies showing that PARP1 facilitates optimal recruitment of FUS to DNA damage sites[17–19] and suggest that FUS may be involved in a repair complex for oxidative DNA damage.

**FUS activates LigIII for DNA ligation via direct interaction**. XRCC1 binds to and stabilizes LigIII, generating the ligation complex that is critical for efficient SSB repair[36], particularly in post-mitotic cells, such as neurons, that lack repair subpathways completed by the replicative DNA ligase, LigI[37,38]. Following our observation that FUS associates with both XRCC1 and LigIII in cellulo, we wondered whether FUS interacts directly with one or both of these proteins. In pull-down assays, PARP1, XRCC1, and LigIII as well as the XRCC1/LigIII complex were specifically retained on glutathione beads liganded by GST-FUS, indicating that FUS interacts with both subunits of the XRCC1/LigIII complex (Fig. 3a). To further evaluate the specificity of FUS–XRCC1 interaction, we performed a broad domain mapping analysis and identified that Glycine-rich region (aa268–aa355) of FUS is the major region involved in XRCC1 binding (Supplementary Fig. 3). The C-terminal aa465–aa526 also exhibited weak binding, while the N-terminal aa1–aa267 did not show detectable binding activity.

To address the functional impact of these interactions, we incubated XRCC1/LigIII with a nick-containing Cy3-labeled duplex oligonucleotide (Fig. 3b, top) prior to separation of the ligated product and unligated substrate oligonucleotides by denaturing gel electrophoresis[39]. FUS significantly stimulated ligation efficiency (Fig. 3b, c) with ~fivefold increase at a 1:2 molar ratio of XRCC1/LigIII to FUS. To further evaluate the enzymatic mechanism of this activation, we analyzed the Michaelis-Menton kinetic parameters and found a threefold decrease in $K_m$ of LigIII but no significant change in $k_{cat}$, in the presence of FUS, indicating enhanced substrate affinity or loading (Fig. 3d–g). Overall, the catalytic efficiency of LigIII was enhanced fourfold by FUS (Fig. 3g).

**DNA ligation defects in CRSPR/Cas9-mediated FUS KO cells**. We next examined the in cellulo impact of FUS on LigIII function by establishing a CRISPR/Cas9-mediated FUS KO HEK293 line (Fig. 4a). After confirming the absence of FUS in extracts from the FUS KO HEK293 derivative by immunoblotting, we confirmed that the steady state levels of XRCC1, LigIII, and PARP1 were similar in the FUS KO and parental cells (Fig. 4b).

Subsequently, we compared the ligation activity of XRCC1 IP complexes isolated from nuclear extracts of parental and FUS KO cells. While the ligation activity of the XRCC1 IP from the FUS KO extract was significantly reduced compared with the XRCC1 IP from the WT HEK293 cell extracts (Fig. 4c, d), the ligation deficiency was rescued by the addition of recombinant FUS to the ligation reaction. This result suggests that the specific ligation defect was caused by FUS deficiency. This is supported by the observation that FUS KO cells have a ~3–5-fold increase in unrepaired DNA strand breaks, as determined by long amplicon (LA)-PCR (Supplementary Figs. 4a, b).

To exclude the possible contribution of non-specific CRISPR/Cas9 targeting in the FUS KO line, we examined the ligation activity of XRCC1 IP complexes from SH-SY5Y cells and its FUS KD derivative using 3′-UTR-specific shRNA (shRNA-2 in Fig. 1a). FUS KD caused a similar reduction in ligation activity compared to FUS KO (Fig. 4c, compare lanes 2 and 3 and 4f, compare lanes 2 and lane 3). In order to attribute the ligation defects to the loss of FUS, we performed complementation analysis with ectopic expression of FLAG-tagged versions of WT FUS and a common and well-characterized familial FUS mutant, P525L, which has a robust nuclear clearance phenotype and is associated with a severe disease progression[40–42]. For this experiment, we constructed Flag-FUS WT and Flag-FUS P525L expression plasmids and ectopically expressed them in FUS-depleted cells using 3′-UTR-specific shRNA. Fluorescence microscopy confirmed the predominant nuclear localization of Flag-FUS WT and significant cytoplasmic localization and low nuclear levels of Flag-FUS P525L (Fig. 4e). While the levels of XRCC1, LigIII, and PARP1 were not affected in FUS KD or ectopically expressing cells (Fig. 4g), the reduced ligation activity of the XRCC1 IP from the KD cells was rescued by expression of WT FUS but not the mutant FUS (Fig. 4f).

Following up on our data demonstrating a specific interaction of FUS with LigIII, but not with LigIV (Fig. 2a) or LigI (Supplementary Fig. 4c), to further confirm the specificity of the functional association of FUS with LigIII, we also measured ligation activity in XRCC1 IP isolated from LigIII KD cells. As expected, there was a significant decrease in DNA nick ligation and addition of recombinant FUS did not enhance the ligation activity in LigIII KD cells (Supplementary Fig. 4d). Altogether, these data clearly show that FUS is required for efficient DNA ligation by LigIII in human cells.

**DNA ligation defects in ALS patients with FUS pathology**. We next investigated whether there was a correlation between FUS pathology and DNA ligation defects in the spinal cord tissue of ALS patients (see Methods for details). Of the 4 control and 10 ALS samples screened by immunoblotting for monomeric vs oligomeric FUS levels, 2 ALS samples (P-6 and P-7) showed >60% reduction in FUS monomer levels (Supplementary Figs. 5a, b). The immunoblots also showed an increase in higher mobility bands corresponding to FUS oligomers. Immunohistochemistry (IHC) of control and ALS spinal cord tissue sections using FUS antibody showed clear evidence for significant cytosolic accumulation of FUS in ALS spinal cord (Supplementary Fig. 5c), similar to the FUS pathology previously demonstrated in ALS-FUS patients[1,2]. Analysis of genomic DNA integrity by LA-PCR detected a ~twofold higher occurrence of strand breaks in P-6 and P-7 ALS spinal cord tissue relative to C-2 and C-3 (Supplementary Fig. 5d). In accord with these observations, DNA ligation capacity in P-6 and P-7 spinal cord tissue extracts was reduced, compared to C-2 and C-3 extracts (Supplementary Figs. 5e, f), even though the levels of XRCC1 and LigIII were comparable in the control and ALS samples (Supplementary

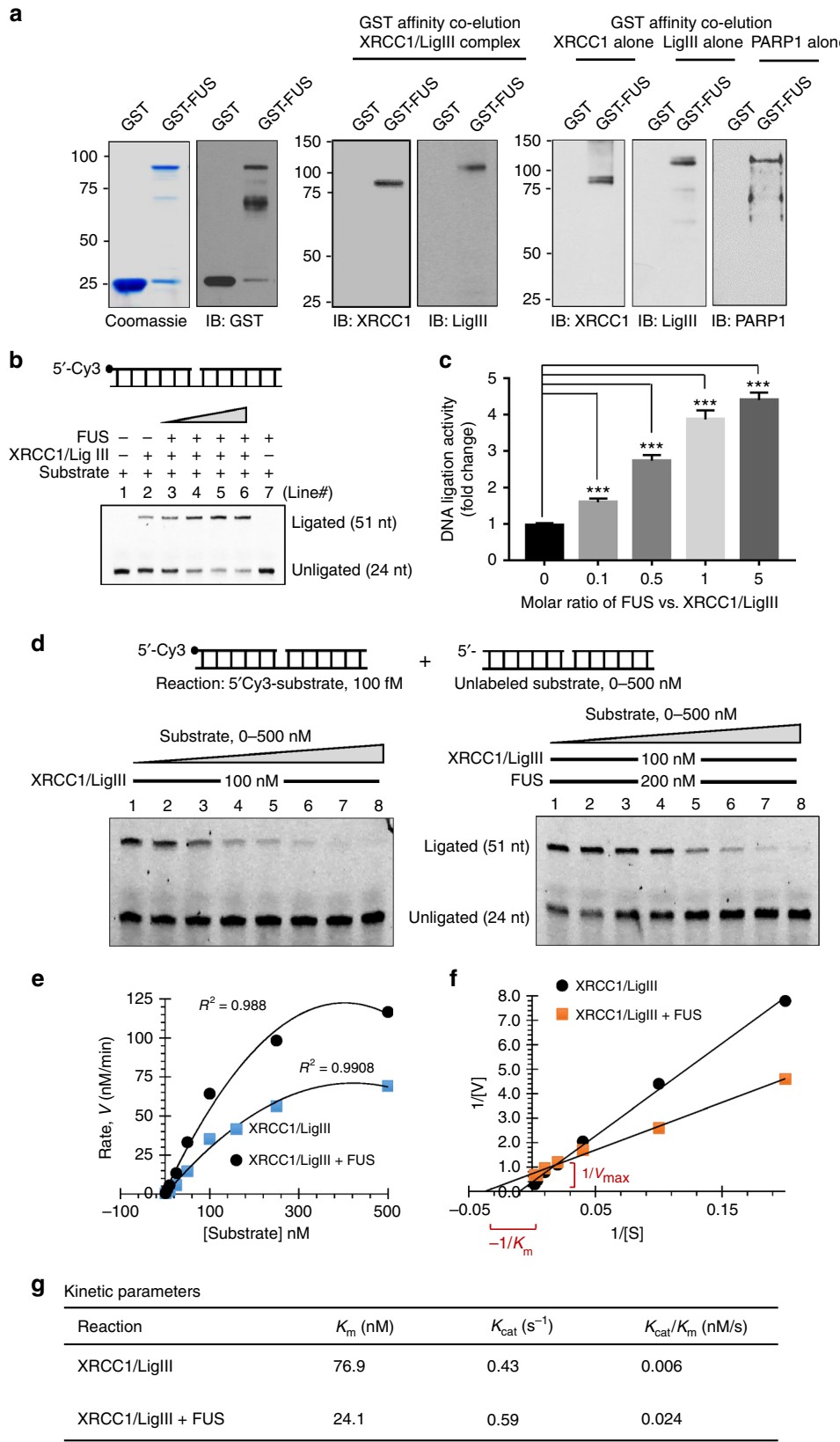

**g** Kinetic parameters

| Reaction | $K_m$ (nM) | $K_{cat}$ (s$^{-1}$) | $K_{cat}/K_m$ (nM/s) |
|---|---|---|---|
| XRCC1/LigIII | 76.9 | 0.43 | 0.006 |
| XRCC1/LigIII + FUS | 24.1 | 0.59 | 0.024 |

Figs. 5g and 5h). The clinical characteristics of all samples are provided in Supplementary Table 1. These data reveal a strong correlation between FUS pathology and DNA ligation defects that is consistent with our cell culture data.

**Defective DNA repair in ALS patient-derived iPSC line.** Several FUS mutations, mostly familial, have been linked to ALS[1,2]. To examine the impact of FUS mutations on oxidative DNA damage repair, we cultured human fibroblast lines and iPSC lines derived

**Fig. 3** FUS activates LigIII for DNA ligation via direct interaction. **a** In vitro affinity co-elution of purified GST-FUS with XRCC1/LigIII complex, or XRCC1, LigIII, and PARP1 separately. GST or GST-FUS was detected by coomassie staining (left panel) or IB. XRCC1, LigIII and PARP1 were detected by IB. **b** and **c**. In vitro DNA nick ligation activity assay. Increasing amounts of FUS (10–500 f.moles) was incubated with XRCC1/LigIII complex (100 f.moles) in a reaction mixture containing 5′ cy3-labeled duplex oligonucleotide substrate carrying a single-strand nick in the middle (top). The reaction was stoped after 30 min and the products were separated by electrophoresis under denaturing conditions. Quantitation of fold change in ligated products (**c**). The error bars are standard deviation of experiments performed in triplicate (***$p < 0.001$, two-tailed unpaired Student's $t$-test). **d–f** Measurement of kinetic parameters of FUS-induced activation of LigIII. Increasing amounts (0–500 nM) of cold (unlabeled) ligation substrate was mixed in a ligation reaction as in Fig. 3b containing XRCC1/LigIII with or without FUS, together with containing 50fM cy3-labeled substrate. The reaction was stopped after 5 min. **e** The plot of the reaction rate (V) vs. substrate concentration. The kinetic parameters $K_m$ and $K_{cat}$ were calculated by Lineweaver–Burk plot of 1/V vs. 1/S (**f**). **g** Tabulation of kinetic parameters. All error bars are standard deviation of three independent experiments

from a healthy control individual and two patients with familial ALS carrying either a R521H or a P525L mutant versions of FUS (Fig. 5a)[35]. We first compared nuclear and cytoplasmic FUS levels in the fibroblasts by immunoblotting. In the FUS-P525L cell line, there was a ~threefold increase in cytoplasmic FUS level and a reduction in nuclear FUS levels relative to the control. In contrast, the R521H cell line had only a small increase in cytoplasmic FUS (Fig. 5b, c). However, both mutant fibroblasts had a comparable increase in the steady state levels of genomic DNA strand breaks, as determined by LA-PCR amplification of a 10.4 kb segment of the *hprt* gene (Fig. 5d, e).

To evaluate the impact of FUS mutations on DNA damage repair in motor neurons, the mutant iPSC lines were induced to differentiate into motor neurons (Fig. 2c)[35]. Motor neuron differentiation efficiency was confirmed by immunofluorescent staining with MAP2, β-tubulin III, and motor neuron-specific markers Isl-1, ChAT (Supplementary Figs. 6a, b). Similar to the observations in the iPSC lines from which they are differentiated, the motor neurons derived from the FUS-P525L mutant line showed significant cytoplasmic FUS accumulation compared to WT FUS motor neurons, while the FUS-R521H motor neurons showed only moderate cytoplasmic accumulation (Fig. 5f). To evaluate SSB repair kinetics in patient-derived motor neurons, cells were treated with GO for 1 h and evaluated 30, 60 or 180 min later. LA-PCR analysis of isolated genomic DNA showed significantly delayed repair of DNA strand breaks in both R521H and P525L mutant motor neurons compared to the control WT cells (Fig. 5g, h). Furthermore, while the untreated mutant FUS motor neurons had only slightly higher numbers of TUNEL-positive cells compared with the WT motor neurons, this difference was increased following incubation with $H_2O_2$, indicating a defect in DNA damage-induced repair (Fig. 5i, j). Furthermore, the alkaline comet assay showed deficient oxidative DNA damage repair in motor neurons with FUS mutations (Supplementary Fig. 6c).

**Correction of FUS mutations rescue DNA ligation defects**. Consistent with the accumulation of DNA damage and delayed DNA repair, the mutant FUS motor neurons exhibited a >50% reduction in DNA nick ligation efficiency, compared with control WT neurons (Fig. 6c, lanes 2, 4, and 6). Again, both R521H and P525L showed comparably impaired ligation efficiency. To directly attribute this DNA ligation defect to the FUS mutations, we corrected the mutation in the iPSC line using the CRISPR/Cas9 knock-in technology. The reversal of R521H to H521R and P525L to L525P was confirmed by sequencing, and we verified the pluripotent self-renewal capacity by embryonic body formation analysis (Fig. 6a; Supplementary Figs. 7a, 7b and 7c)[35]. The origin of the isogenic controls was confirmed by SNP analysis (Supplementary Table 2)[35]. These isogenic control lines were differentiated to motor neurons and FUS distribution and DNA ligation capacity were examined. AS expected, correcting FUS mutations rescued the nuclear FUS clearance phenotype in iPSCs

(Supplementary Fig. 7d) and motor neurons (Fig. 6b). Furthermore, as shown in Fig. 6c, d, the correction of the FUS mutation completely rescued the ligation defect. To further test whether the mutation correction reverted the delayed repair of DNA strand breaks, we performed LA-PCR-based DNA integrity measurement at early (30 min) and late (180 min) time points after GO treatment (Fig. 6e). The data at 30 min showed a comparable level of DNA damage induced by GO. At 180 min, DNA integrity was mostly restored in mutation-corrected cells (~90% DNA integrity), whereas the mutant cells still showed significantly reduced (~60%) DNA integrity, confirming that the observed ligation defect and delayed repair are indeed caused by the FUS mutations. Altogether, these results confirm that the R521H and P525L mutant versions of FUS fail to enhance nick ligation, leading to the accumulation of unrepaired DNA strand breaks in motor neurons.

**Dominant negative activity of the R521H FUS mutation**. While the ligation defect in mutant FUS-P525L cells can be attributed to the increased nuclear clearance of the mutant FUS protein, nuclear level of the FUS-R521H mutant was only slightly reduced. This suggests that the reduced DNA ligation in cells expressing FUS-R521H occurs by a different mechanism. To address this question, we examined the DNA damage recruitment and repair complex formation of FUS-R521H and FUS-R521C mutants. GFP-tagged WT and mutant FUS proteins were ectopically expressed in HEK293 cells. While GFP-FUS-WT and GFP-FUS-R521H were predominantly localized in the nucleus (Fig. 7c), the FUS-R521C mutant exhibited a few clear cytoplasmic aggregates (Fig. 7c). Recruitment of the GFP fusion proteins to MIR-induced DNA damage tracks was monitored in live cells. A 365 nm low-intensity laser was used to generate predominantly oxidative damage in the laser track[20,43]. Both FUS mutants showed significantly reduced recruitment at the damage track (Fig. 7a, b, Supplementary Movie 1, 2 and 3).

We next examined the association of WT and FUS-R521H proteins with SSB repair proteins using the PLA. Flag-tagged WT and mutant FUS proteins were expressed in SH-SY5Y cells at comparable levels (Supplementary Fig. 8a). PLA of Flag vs XRCC1, LigIII, or PARP-1 in GO-treated SH-SY5Y cells, showed a significantly reduced association (>80 %) of mutant FUS with XRCC1, LigIII, and PARP-1 compared with WT FUS (Supplementary Figs. 8b, c). The reduced association of FUS-R521H with SSB proteins was also observed in iPSCs (Fig. 7d, e). Thus, the FUS-521H mutant is defective in recruitment to damage sites and repair complex formation despite being present in the nucleus.

Two independent approaches were employed to determine whether the effect of the FUS-R521H allele is solely due to haploinsufficiency of functional FUS or a dominant negative activity of the FUS-R521H protein. First, we optimized ~50% transient KD of FUS in WT human motor neurons using antisense oligonucleotides (ASOs) and measured the LigIII

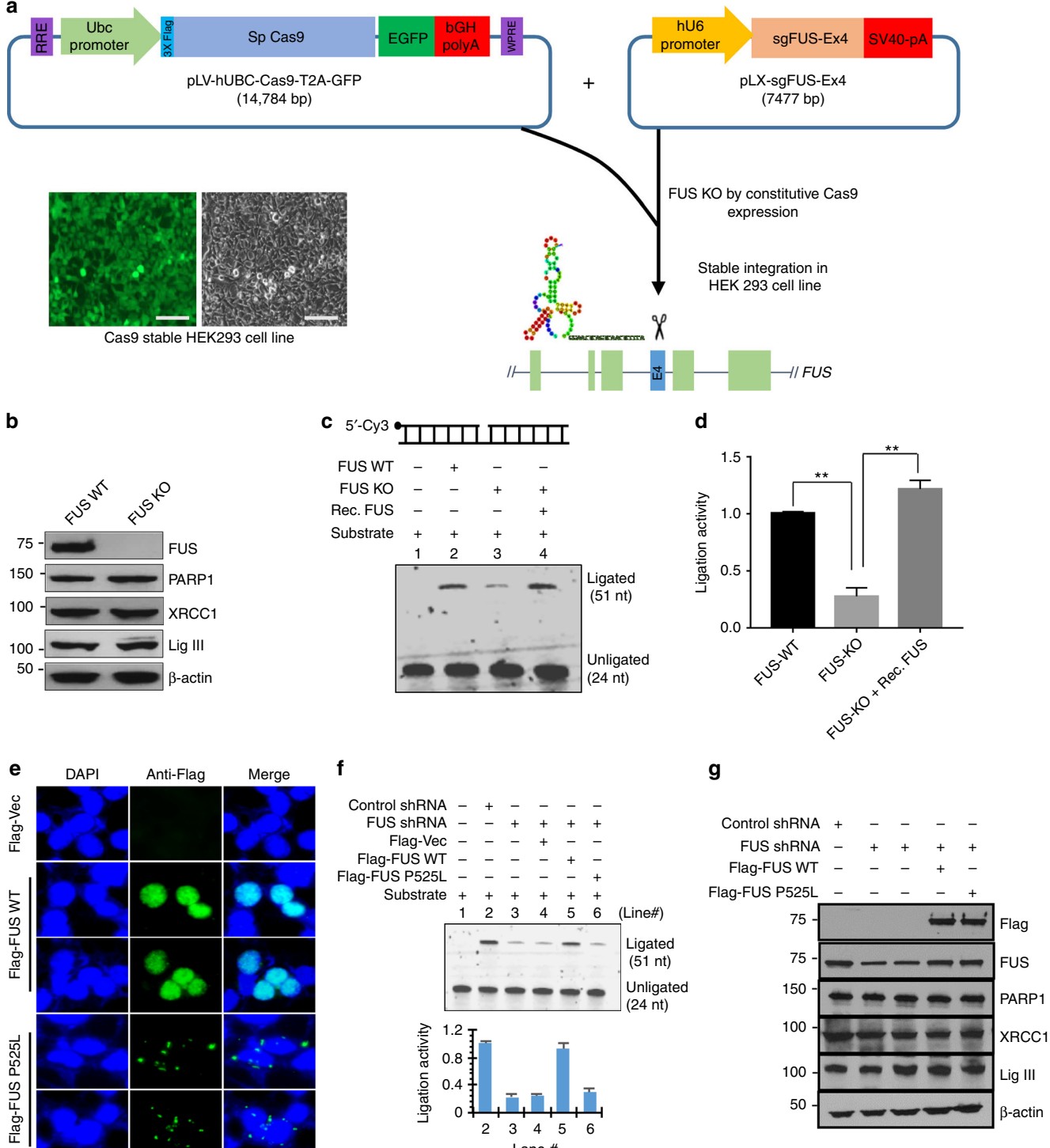

**Fig. 4** DNA ligation defects in CRSPR/Cas9-mediated FUS KO cells. **a** Scheme of CRISPR/Cas9 targeting of FUS. Images indicate stable expression of GFP-Cas9 in HEK293 cells. Scale bar = 50 μm. **b** IB of endogenous FUS, PARP1, XRCC1 and LigIII in FUS WT and FUS KO HEK293 cells. β-actin was probed as loading control. **c, d** In vitro DNA nick ligation activity assay. XRCC1 IP complex from nuclear extract of GO-treated FUS WT and FUS KO HEK293 cells, was incubated with or without purified FUS. **d** Quantitation of ligation activity (**p < 0.01, two-tailed unpaired Student's t-test). **e** IF of ectopically expressed Flag-FUS WT and Flag-FUS P525L mutant in HEK293 cells. Endogenous FUS depleted using UTR-specific shRNA. IF was performed by anti-Flag antibody 48 h after the co-transfection of FUS shRNA with either Flag-FUS WT or Flag-FUS P525L plasmid. DAPI staining indicates nucleus. Scale bar = 5 μm. **f** In vitro DNA nick ligation assay. Rescue of DNA ligation defect in FUS KD HEK293 cells by WT vs P525L mutant FUS expression. XRCC1 IP complex from nuclear extract of GO-treated FUS KD HEK293 cells with or without ectopic FUS WT and FUS P525L expression. The quantitation of ligation activity is shown in the histogram (bottom). **g** IB of cell extract used in **f** to confirm the comparable level of PARP1, XRCC1, and LigIII in WT or mutant FUS expressing cells. β-actin was probed as loading control. All error bars are standard deviation of experiments performed in triplicate

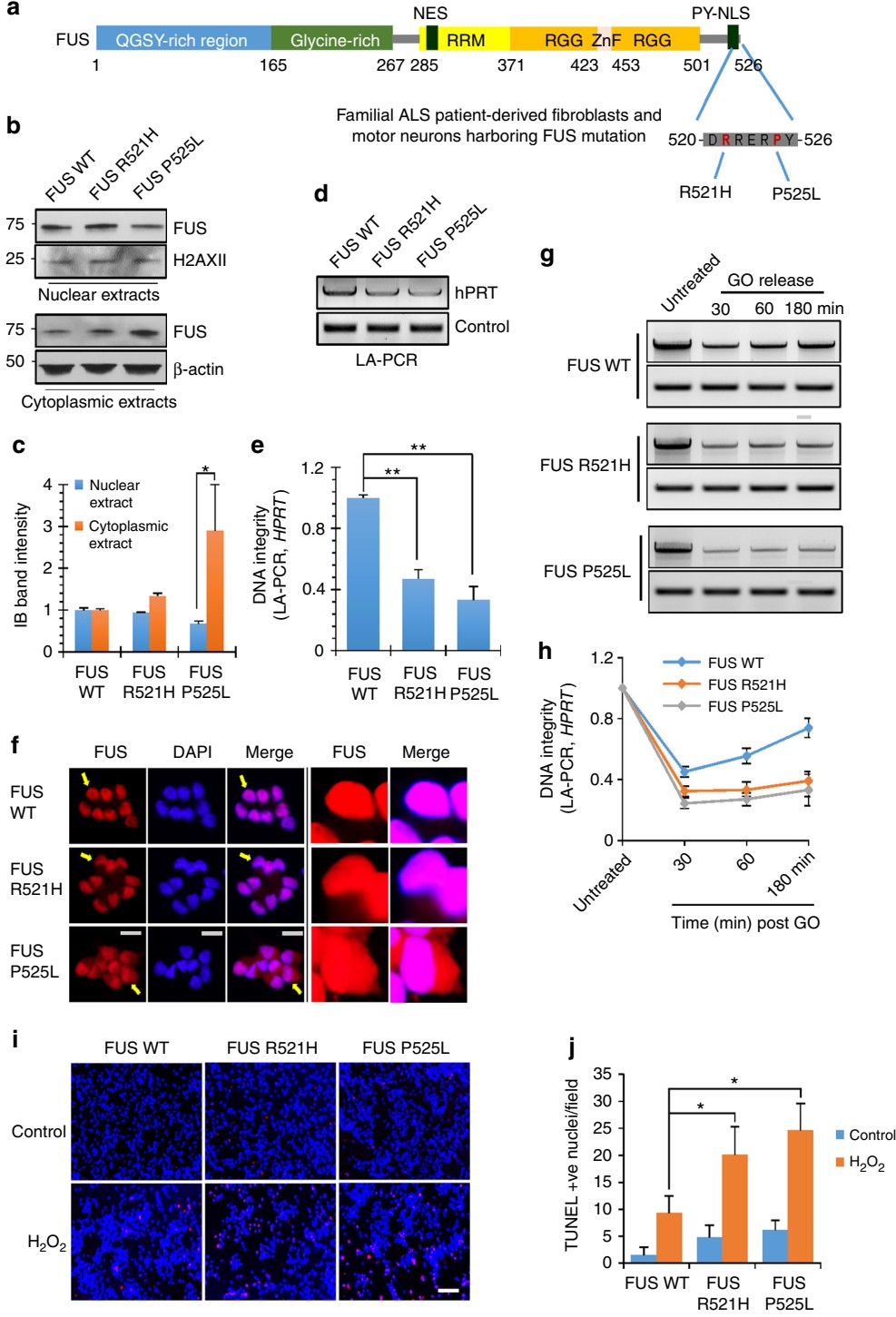

activity in XRCC1 IP isolated from control vs 50% FUS KD cells. 50% KD was confirmed by both mRNA quantification (Fig. 7f) and immunoblotting of protein level (Fig. 7g, h). Incubation with 15 nM ASOs for one week caused ~50% KD of FUS in motor neurons. The 50% FUS KD cells showed a moderate (~20%) reduction in LigIII activity (Fig. 7i, j; Lane 1 vs 2). We then compared the LigIII activity in the patient-derived R521H mutant FUS cell line with the FUS KD line. Theoretically, the heterozygous mutant line derived from an ALS patient is expected to have 50% WT and 50% mutant FUS. However, the ligase activity in mutant motor neurons was significantly lower

compared to both WT and 50% KD lines. Quantitation of the ligase activity from three independent experiments showed that the ligase activity in the mutant cells was ~50% lower compared to WT cells and it was reduced an additional ~30% compared to FUS KD cells (Fig. 7j). These data are in line with a dominant negative effect of mutant FUS on the LigIII activity, rather than with only haploinsufficiency.

In a complementary approach, we used cell lines that ectopically express WT and mutant FUS at a comparable level (~2-fold greater than endogenous) following induction with doxycycline (Supplementary Figs. 8d, e) and measured

**Fig. 5** Defective DNA repair in ALS patient-derived iPSC line. **a** The location of familial FUS mutations R521H and P525L indicated in FUS protein sequence. **b**, **c** IB of nuclear and cytosolic extracts isolated from ALS patient-derived fibroblasts, probed for FUS. H2AXII and β-actin were probed as loading control. Histogram shows quantitation of IB band intensity. The error bars are standard deviation of experiments performed in triplicate (*$p < 0.05$, two-tailed unpaired Student's $t$-test). **d**, **e** Integrity of genomic DNA isolated from ALS patient-derived fibroblasts measured by long amplicon quantitative PCR (LA-PCR) analysis. 10.4 kb fragment including exons 2–5 of the *hprt* gene was amplified and separated in 1% agarose gel. The amplified DNA product was quantified using pico green fluorescence. The error bars are standard deviation of experiments performed in triplicate (**$p < 0.01$, two-tailed unpaired Student's $t$-test). **f** IF motor neurons for FUS to analyze its nucleus vs. cytosolic distribution. Nuclei stained with DAPI. Zoomed images on right showing FUS distribution in arrow indicated nuclear. Scale bar = 10 μm. **g**, **h** LA-PCR-based DNA damage repair kinetic analysis. Genome DNA extracted from iPSC-derived motor neurons at indicated time points after release from exposure to GO (100 ng/ml) for 1 h. Amplification products analyzed by agarose gel electrophoresis or pico green-based quantitation as in Fig. 6d,e. The error bars are standard deviation of experiments performed in triplicate. **i**, **j** TUNEL analysis of motor neurons carrying FUS WT or FUS R521H or P525L mutation. Motor neurons differentiated from ALS patients derived iPSC lines were treated with H$_2$O$_2$ (100 μM for 3 h), and cells were subjected to TUNEL assay 24 h after the release from treatment. Scale bar = 50 μm. Histogram shows quantitation of TUNEL-positive nuclei from 200 cells in total. The error bars are standard deviation (*$p < 0.05$, two-tailed unpaired Student's $t$-test)

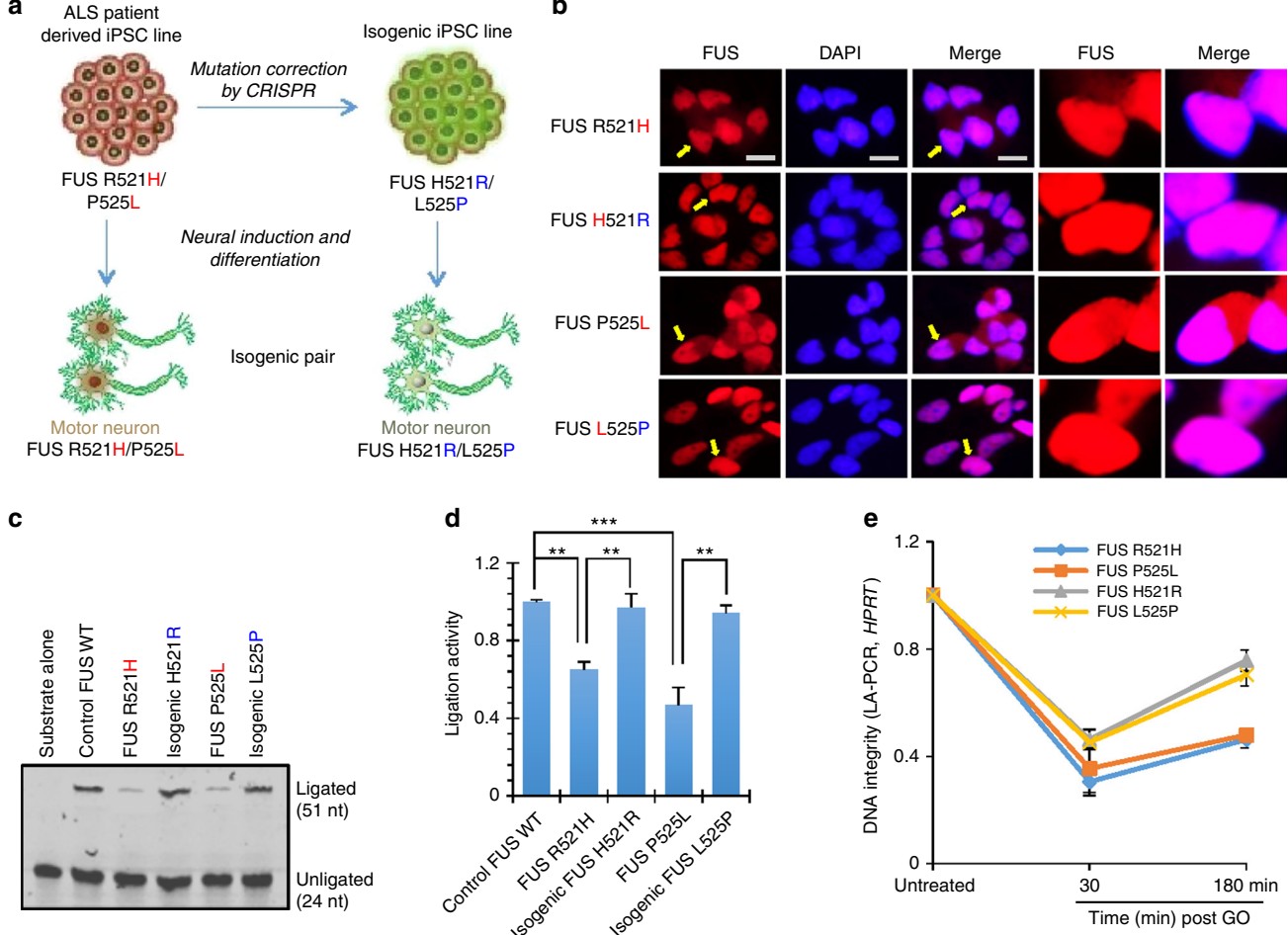

**Fig. 6** Correction of FUS mutations rescue DNA ligation defects. **a** Scheme of correction of FUS mutation in familial ALS patient-derived iPSC lines by CRISPR knock-in to generate isogenic control lines and their differentiation to motor neurons. **b** IF motor neurons for FUS to analyze its nucleus vs. cytosolic distribution. Nuclei stained with DAPI. Zoomed images on right showing FUS distribution in arrow indicated nuclear. Scale bar = 10 μm. **c**, **d** In vitro DNA ligation assay. XRCC1 IP complex from nuclear extract of mutant and isogenic control motor neurons and quantified (**$p < 0.01$; ***$p < 0.001$, two-tailed unpaired Student's $t$-test). **e** LA-PCR-based DNA damage repair kinetic analysis. Genome DNA extracted from iPSC-derived motor neurons at indicated time points after release from exposure to GO (100 ng/ml) for 1 h. Amplification products analyzed by pico green-based quantitation. All error bars are standard deviation of experiments performed in triplicate

LigIII activity in XRCC1 IPs. The inducible human H9 embryonic stem cells (hH9-ESCs) contained WT and R521H mutant FUS in a safe harbor locus[35]. Expression of WT FUS increased LigIII activity by ~20% whereas expression of FUS R521H) decreased ligation activity by about 25% compared to control cells (Supplementary Figs. 8f, b). This clearly shows that mutant FUS can reduce the LigIII activity, despite the fact that endogenous FUS was present. The ability of FUS R521H to reduce ligation activity in the presence of comparable levels of endogenous WT FUS protein indicates that the mutant FUS

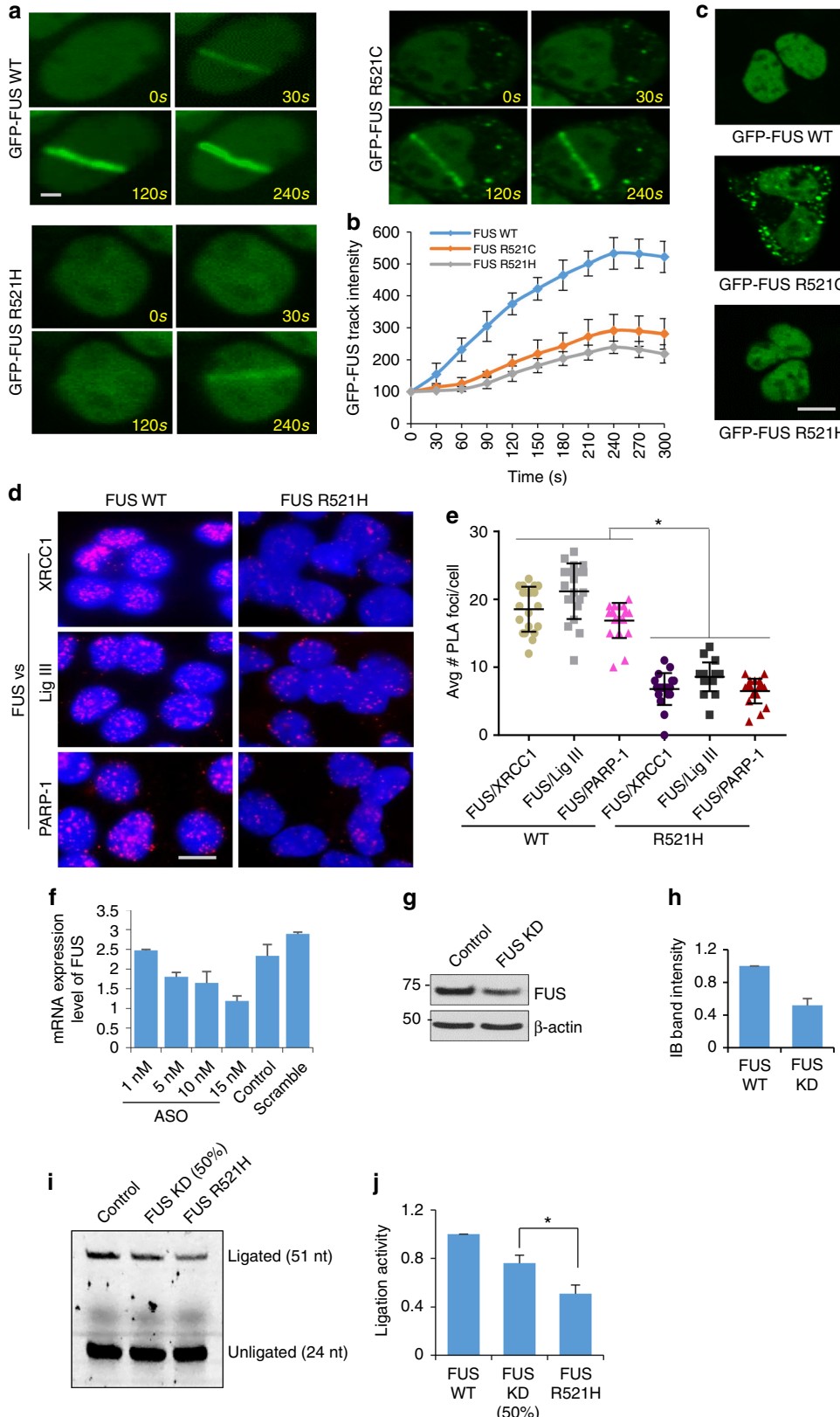

reduces LigIII-dependent ligation, at least in part, by acting in a dominant negative manner.

**FUS facilitates PARP-1-dependent recruitment of XRCC1/LigIII.** FUS was previously shown to be rapidly recruited to UVA irradiation-induced DNA damage tracks in a PARP-dependent fashion[17–19]. PARP1 also facilitates recruitment of XRCC1/LigIII to damage sites[28–30]. To further dissect the role of FUS in this early oxidative DNA damage response, we examined the effect of FUS depletion on XRCC1 recruitment to MIR-induced DNA damage tracks. GFP-XRCC1, which was ectopically expressed in

**Fig. 7** Dominant negative activity of the R521H FUS mutation. **a**, **b** Recruitment of GFP-FUS at laser ablation track by live cell imaging. Representative images of ectopic GFP-FUS WT or R521H/C mutant following laser ablation. GFP-FUS WT or R521H/C was transfected into HEK293 cells and subjected to laser ablation 48 h after the transfection. The recruitment of GFP fluorescence at laser tracks was monitored by live cell imaging (see movie clip in Supplementary Movie 1, 2 and 3 for live imaging). Scale bar = 1 μm. **b** Quantification of GFP-FUS track intensity from 15 cells is shown in histogram. The error bars are standard deviation. **c** Representative IF images of GFP-FUS WT or R521H/C mutant in HEK293 cells without laser treatment. Scale bar = 5 μm. **d**, **e** PLA of FUS vs XRCC1, LigIII and PARP1 in iPSC carrying FUS WT or FUS R521H after GO treatment. Representative images shown in **d** and average number of PLA foci from 25 cells were quantified in **e**. The error bars are standard deviation (*$p < 0.05$, two-tailed unpaired Student's $t$-test). Scale bar = 5 μm. **f** Quantification of FUS mRNA analyzed by RT-PCR in antisense oligonucleotide-incubated motor neurons at indicted concentrations. The error bars are standard deviation of experiments performed in triplicate. **g**, **h** IB showing FUS KD in **f** Histogram shows quantitation of IB band intensity (**h**). The error bars are standard deviation of experiments performed in triplicate. **i**, **j** In vitro DNA nick ligation activity assay. XRCC1 IP complex from GO-treated motor neurons with FUS WT, FUS KD, and FUS R521H, and quantified. The error bars are standard deviation of experiments performed in triplicate (*$p < 0.05$, two-tailed unpaired Student's $t$-test)

HEK293 cells, was rapidly recruited to the laser track within 30 s and was retained until 300 s as expected (Fig. 8a; Supplementary Movie 4). Depletion of FUS by shRNA significantly delayed XRCC1 recruitment and reduced the total amount of XRCC1 that was recruited (Fig. 8a, b; Supplementary Movie 5).

In addition, there was a marked reduction in the oxidative stress-dependent association of PARP-1 and XRCC1 in FUS KD cells measured by PLA (Fig. 8c, d) and reduced levels of LigIII and XRCC1 in PARP1 IP from fibroblasts expressing either FUS-R521H or FUS-P525L (Fig. 8e, f). To test whether the FUS–XRCC1 interaction is modulated by PARylation, we examined the proteins co-immunoprecipitated by a FUS antibody from extracts cells treated with or without the PARP1 inhibitor, AG-14361 (Supplementary Fig. 9a). The levels of both XRCC1 and LigIII were markedly reduced in the FUS IPs from AG-14361-treated cells. Similarly, the PLA signals for FUS vs PARP-1 were reduced in AG-14361-treated cells (Supplementary Fig. 9b). These data suggest that although FUS interacts directly with XRCC1 in a binary fashion in vitro, the interaction is promoted by PARP-1 activity in cells. To investigate this further, we performed an in vitro ADP-ribosylation assay using purified PARP1 protein together with NAD$^+$ and octameric oligonucleotide as described previously[44]. PARylated PARP1 was detected by immunoblotting with PAR antibody (Supplementary Fig. 9c). Surprisingly, when we added purified FUS protein to the reaction, the auto-PARylation level of PARP-1 was increased by ~10-fold. The PAR antibody detected a second lower mobility band that corresponded in size to FUS, suggesting that FUS may be PARylated by PARP-1 in vitro. To test the effect of PARP-1 activity on the FUS–XRCC1 interaction, we performed GST affinity pull-down in the presence of PARP-1 and NAD$^+$ and found that PARylation enhanced the in vitro interaction of FUS and XRCC1 (Supplementary Fig. 9d). Together, these data show that PARP-1 and its PARylation activity enhance the interaction of FUS with XRCC1, which is critical for the recruitment of XRCC1/LigIII at oxidatively damaged genomic DNA (schematically shown in the Fig. 8g).

## Discussion

In this study, we identified FUS as a critical component of the oxidative genome damage repair complex. ROS generate SSBs both directly and also indirectly during the repair of oxidized bases by BER[26]. PARP1 acts as the SSB sensor that recruits other SSBR proteins, including XRCC1/LigIII, in a PARylation-dependent manner[26,28,29]. The basic BER/SSBR pathway involves four key reactions: (1) excision of the base lesion by a DNA glycosylase, (2) end-processing at the SSBs to generate 3′ OH and 5′-phosphate ends, compatible with gap-filling synthesis and ligation, (3) gap-filling by a DNA polymerase, and (4) final nick sealing by a DNA ligase[33]. Nick sealing is the critical rate-

limiting step in both BER and SSBR. In contrast to cycling cells that utilize both LigI and XRCC1/LigIII to complete BER and SSBR, XRCC1/LigIII is the predominant activity in post-mitotic cells such as motor neurons[45]. Our comprehensive in cellulo and in vitro studies reveal stable and direct interaction between FUS and XRCC1/LigIII that enhances DNA nick ligation to protect the genome from oxidative damage and show that loss of FUS function results in DNA nick ligation defects, significantly reduced SSB repair efficiency, and cellular vulnerability to oxidative insults.

Our data in fibroblasts obtained from familial ALS patients with the R521H and P525L FUS mutations, and iPSCs/motor neurons derived from these fibroblasts indicate that the two familial mutants, R521H and P525L, cause ligation defects by distinct mechanisms: the DNA damage-dependent association of nuclear FUS-R521H with PARP1 and XRCC1/LigIII is reduced, whereas FUS-P525L nuclear levels are low due to its aberrant cytoplasmic localization. These cells showed a significant defect in DNA ligation, which was rescued by addition of recombinant FUS or by correcting the genomic FUS mutations. Furthermore, the R521H mutant inhibited LigIII-dependent joining activity in a dominant negative fashion. The dominant nature of the toxic gain-of-function of FUS mutants observed in our study is consistent with recent FUS KO and mutant transgenic mice studies. While a heterozygous FUS KO mice was viable and did not develop strong ALS-like phenotype[46], the expression of a mutant FUS transgene induced selective motor neuron degeneration in mice[47].

Our study provides specific molecular insights into a previously undescribed DNA repair defect linked with FUS-associated neurodegeneration. It was reported that PARP is involved in forming liquid compartments of FUS at DNA damage sites, and aberrant phase transition of the liquid to solid-like FUS aggregates could be involved in the disease onset[48]. Other studies showed recruitment of FUS to DNA damage sites in a PARP1-dependent manner via its affinity for PAR[17,18]. Our data show that once FUS is recruited at damage sites it facilitates the recruitment of XRCC1/LigIII. Interestingly, we observed a direct interaction between FUS and PARP-1 in vitro. Although FUS was previously shown to bind to PAR groups[17,18] its direct binding with PARP-1 has never been shown. Our study thus documents a direct binding of FUS to PARP-1, XRCC1 and LigIII, which is enhanced by the PARylation activity of PARP-1. It is likely that PAR on auto-PARylated PARP-1 provides the initial signal for the recruitment of FUS–XRCC1-LigIII to the PARP-1-bound-damage sites, with direct binding stabilizing these interactions. Our in vitro PARylation data also indicate that FUS may be PARylated by PARP-1, consistent with previous observation of mass spec screening[49], whose functional role in genome maintenance needs to be investigated.

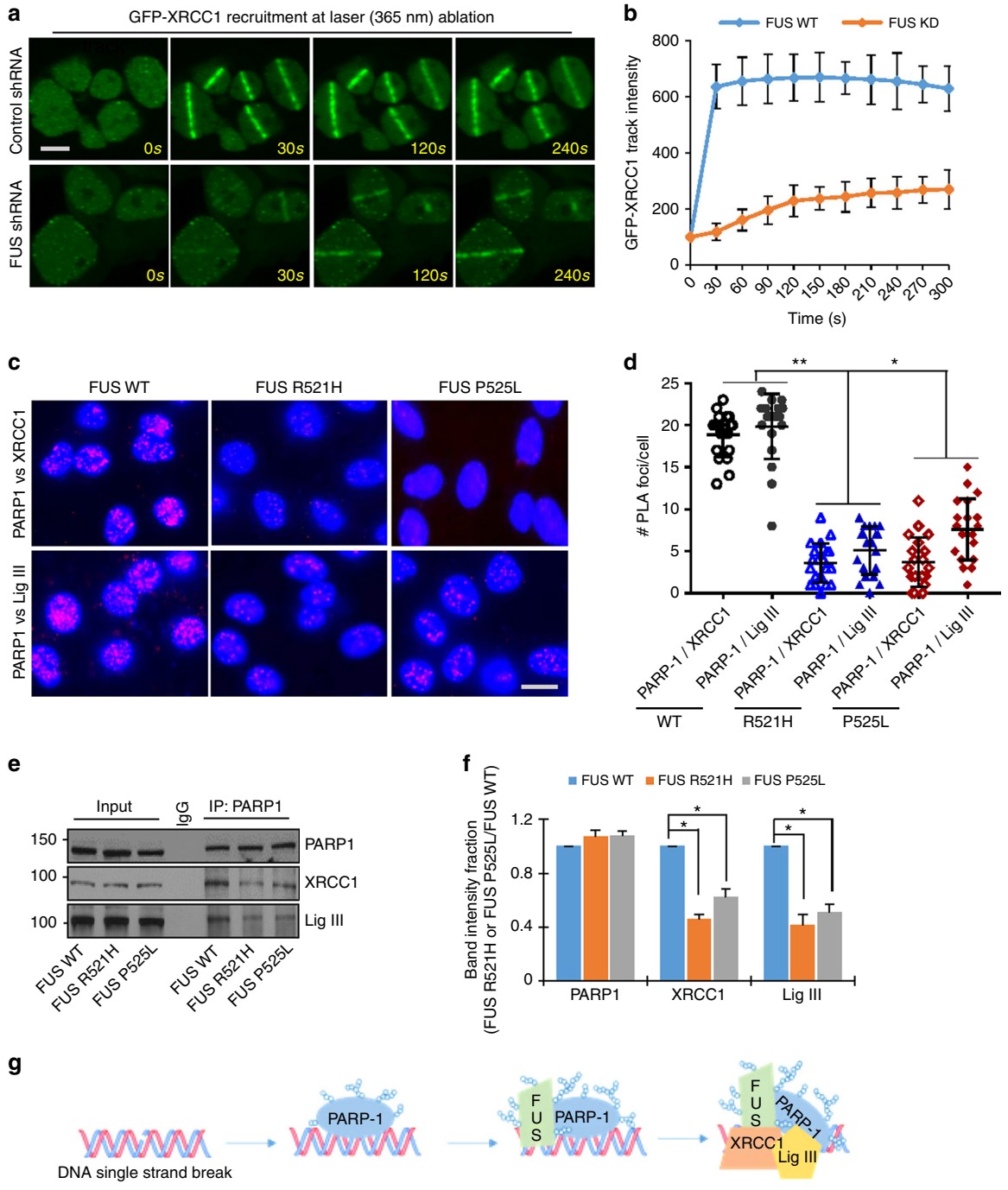

**Fig. 8** FUS facilitates PARP-1-dependent recruitment of XRCC1/LigIII. **a, b** Laser ablation microscopy for recruitment of GFP-XRCC1 after FUS knockdown. GFP-XRCC1 and FUS shRNA were co-transfected into HEK293 cells followed by laser treatment and microscopy 48 h after the transfection, as in Fig. 7a. Scale bar = 5 μm. Histogram shows the intensity of GFP-XRCC1 at laser track (see movie clip in Supplementary Movie 4 and 5 for live imaging). The error bars are standard deviation. **c, d** PLA of PARP1 vs XRCC1 and PARP-1 vs LigIII in WT or FUS mutant patient-derived motor neurons after GO treatment. Nuclei stained with DAPI. The average number of PLA foci from 25 cells were quantified. The error bars are standard deviation (*p < 0.05; **p < 0.01, two-tailed unpaired Student's t-test). Scale bar = 5 μm. **e, f** XRCC1 co-IP (endogenous) from WT or FUS mutant patient-derived fibroblast lines and IB probed for PARP-1 and XRCC1. Histogram shows quantitation of IB band intensity. The error bars are standard deviation of experiments performed in triplicate (*p < 0.05, two-tailed unpaired Student's t-test). **g** A model showing that PARP-1-dependent recruitment of XRCC1/LigIII at DNA SSBs is facilitated by FUS

To further evaluate the biological implications of these findings in ALS patients, we analyzed spinal cord tissue of sporadic ALS patients. The increased DNA damage together with reduced DNA ligation activity in these tissues broadly correlated with FUS pathology shown in immunoblotting and IHC studies, supporting our in vitro findings. Together these data suggest that low LigIII activity and SSB repair defects may be a common pathological mechanism of FUS-dependent ALS (schematically represented in Fig. 9).

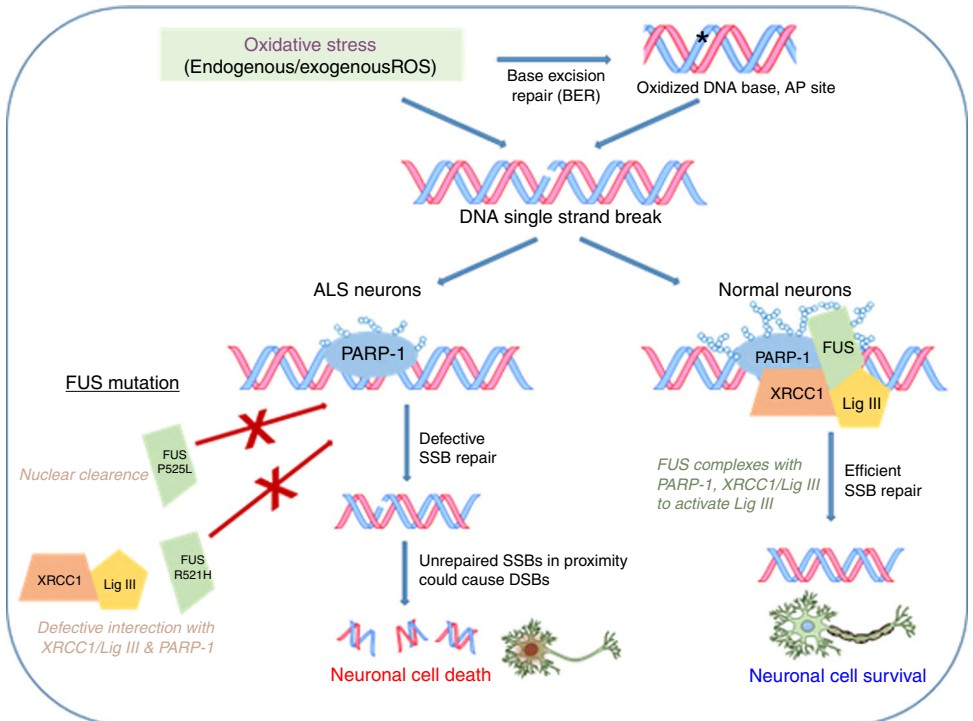

**Fig. 9** Mutant FUS induces DNA ligation defects to inhibit oxidized damage repair. A model showing the involvement of FUS for optimal DNA nick ligation in healthy neurons to facilitate efficient oxidative genome damage repair, and how loss of functional FUS in ALS leads to DNA nick ligation defects. FUS is required for PARP-1-dependent recruitment of XRCC1/LigIII complex at oxidative DNA damage sites. Two familial FUS mutations impair DNA nick ligation by distinct mechanisms. Substantial cytoplasmic localization of P525L FUS causes DNA ligation defect due to loss of functional FUS from nucleus. The R521H/C FUS fails to form the repair complex with XRCC1/LigIII and PARP-1 and is not recruited at damage site. Unrepaired DNA SSBs caused by impaired DNA nick ligation, together with secondary DSBs that may be generated from closely placed single-strand breaks, may substantially contribute to neurodegeneration in familial FUS-ALS patients

The hypothesis of defective DNA repair in ALS was postulated as early as 1982 by Bradley et al., who proposed that abnormal DNA in ALS may arise from deficiency of an isozyme of a DNA repair enzyme[50]. Subsequent studies observed abnormal activity of DNA repair components, including APE1, DNA glycosylase OGG1, mitochondrial DNA polymerase γ, and PARP1, in ALS patients or mouse models, strongly supporting the model for impaired oxidative DNA damage repair in ALS. In line with our findings, a recent study reported that FUS pathology relies on PARP, which can cause axonal degeneration in ALS patient-derived motor neurons[16].

Notably, while previous studies suggested a role of FUS in the repair of DSBs, the relative presence of DSB vs non-DSB damage caused by loss of FUS or FUS mutations was not known. One study[17] showed co-localization of FUS with γ-H2AX at laser (405 nm) ablation sites to suggest its possible presence at DSB sites. Another report[18] showed the presence of FUS at UVA (351 nm) damage sites to suggest its linkage to oxidative damage. It is important to note that laser ablation does not specifically induce a single type of damage; rather, it forms both DSB and non-DSB damage in proximity to laser track. Thus, accurately attributing the co-localization data to a specific type of damage is not possible. Our alkaline and comet assay data demonstrate that loss of FUS predominantly caused SSBs, rather than DSBs. A small fraction of DSBs may be generated secondarily from closely spaced bi-stranded SSBs and from repair intermediates of other oxidative lesions[31]. This result was consistent with our previous observation[51] that neurodegenerative brain tissue accumulated significantly more SSBs than DSBs. Since the brain is generally protected by the blood–brain barrier, endogenous ROS-induced

genome damage is likely the most critical threat to neuronal cells[52]. The major lesions induced by ROS include oxidized bases/sugar fragments, AP sites, and SSBs. Most of these lesions are repaired by the BER and SSBR pathways, which are largely dependent on LigIII in non-dividing and terminally differentiated cells. The loss of nuclear LigIII function may have more profound effect in post-mitotic cells unlike in cycling cells, due to lack of back up ligases in neurons. LigI, which is primarily involved in replication-associated LP-BER, and whose level is very low in post-mitotic cells[53] did not, in contrast to LigIII, specifically associate with FUS. Consistently, our data showed specific association of FUS with LigIII but not with LigI or LigIV. Thus, defects in LigIII function caused by FUS abnormalities are likely to contribute to genomic instability and neuronal cell death in ALS. It is also reasonable to speculate that while loss of a small fraction of spinal cord neurons in the CNS results in motor phenotype, other tissues may be more tolerant to loss/dysfunction of a small fraction of cells.

It is important to note that our results describing a specific mechanism of defective repair of oxidative damage that is FUS dependent, adds to the growing evidence that DNA repair defects play a major role in neurodegeneration[15,54], including hereditary syndromes caused by mutations in PNKP, Aprataxin and TDP1 all of which are involved in the repair of oxidative damage and SSBR[33]. Furthermore, the observations of increased oxidative damage and reduced repair in Alzheimer's disease are consistent with the linkage of defective repair of oxidative genome damage with neurodegeneration[33].

Finally, although our understanding of the pathological and biochemical changes in ALS has increased, there is still no cure.

Currently available treatments only temporarily slow disease progression and do not prevent neuronal death. This short-coming underscores the need for a mechanism-driven approach to effectively prevent onset and delay progression. The identification of a defect in DNA nick ligation that likely contributes to the pathological changes in FUS-linked ALS opens avenues for intervention strategies for ALS and other FUS-linked neurodegenerative disease that enhance DNA LigIII activity and/or DNA repair.

## Methods

**Cell lines, cell culture, and tissue origin**. Human neuroblastoma SH-SY5Y (ATCC) and embryonic kidney HEK293 (ATCC) cell lines were grown in Dulbecco's modified Eagle's medium (DMEM)/F12 or DMEM (Hyclone), respectively, with 10% fetal bovine serum (Hyclone) and 100 U/ml each of penicillin and streptomycin (Hyclone). Human iPSCs (ATCC and VIB-KU Leuven[35]) were maintained on Geltrex LDEV-Free, hESC-Qualified, Reduced Growth Factor Basement Membrane Matrix (GibcoTM) in Essential8TM medium (GibcoTM)[35]. Human fibroblasts (VIB-KU Leuven[35]) were grown in DMEM/F12 medium containing 10% fetal bovine serum, 1% MEM non-essential amino acids (Gibco), and 1.6% Sodium Bicarbonate (Corning). All cells were cultured in an incubator at 37 °C in 5% $CO_2$.

Human ALS and matched control spinal cord tissues were obtained as de-identified specimen from the Department of Veteran's Affairs (VA) Biorepository, USA. Studies on human tissues were conducted in accordance with the ethics board standards at the Department of Veteran's Affairs (VA) and the institutional review boards at the Houston Methodist Research Institute (Houston, Texas).

Primary human fibroblasts were obtained from skin biopsies of ALS patients and controls with the approval of the ethical committee of the University Hospitals Leuven. All other the cell lines (original source: ATCC) were routinely analyzed by PCR for mycoplasma contamination.

**Antibodies, plasmids, shRNAs, and siRNAs**. Rabbit anti-FUS (Cat# A300–302A) antibody was purchased from Bethyl Laboratories, Inc. Rabbit anti-PARP1 antibody (Cat# sc-25780) was purchased from Santa Cruz. Rabbit anti-XRCC1 antibody (Cat# ab134056) was purchased from Abcam. Mouse anti-FLAG antibody (A8592) was purchased from Sigma-Aldrich. Mouse anti-XRCC1 antibody (Cat# TA500880) was purchased from Origene. Mouse anti-LigIII antibody (Cat# ab587) was purchased from Abcam. Mouse anti-Poly (ADP-Ribose) Polymer antibody (Cat# ab 14459) was purchased from Abcam. Fluorescent secondary antibodies, Alexa Fluor 488 anti-mouse (Cat# A28175), and Texas Red anti-rabbit antibody (Cat# T-2767) were purchased from Life Technologies. The concerned antibodies were diluted at 1:1000 for western blotting, 1:500 for immunofluorescence and 1:100 for PLA. GST-FUS plasmid (pGEX6P-1) and GST-FUS domain polypeptide-expression plasmids for aa266–526, aa356–526, and aa465–526 were purchased from Addgene. The aa1–267 FUS polypeptide was cloned into pGEX6P-1 vector as per standard protocol. FUS coding sequence from GST-FUS plasmid was re-cloned in to pCDNA3.1 vector as a C-terminal Flag-FUS construct, and Flag-FUS-R521H, Flag-FUS-P525L, GFP-FUS-R521C, and GFP-FUS-P525L mutants were then generated using a QuickChange II XL Site-Directed Mutagenesis kit (Agilent Technologies), following the manufacturer's instructions. GFP-FUS and GFP-XRCC1 were gifts from Dr. Lawrence J. Hayward (University of Massachusetts Medical School, Worcester, MA.) and Dr. Li Lan (University of Pittsburgh, Pittsburgh, PA), respectively. FUS shRNA plasmids were purchased from Sigma. XRCC1 siRNA was purchased from Dharmacon and LigIII siRNA was purchased from Sigma[55].

**Human spinal cord total tissue extract preparation**. Human post mortem spinal cord tissue of cervical region from ALS patients and age-matched controls were obtained as de-identified specimen from the Department of Veteran's Affairs (VA) Biorepository, USA. The patient clinical features are listed in Supplementary Table 1. Tissues were homogenized in RIPA buffer with protease and phosphatase inhibitors (20 mM Tris-HCl, pH 7.5; 150 mM NaCl; 1 mM EDTA, 0.5 EGTA; 1% Sodium deoxycholate; 1% Tritonx-100; and 0.1% SDS). Tissue lysates were sonicated at amplitude 8 for 10 s, 6–7 times, with 2 min intervals between two consecutive pulses. This was followed by centrifugation at $16,000 \times g$ for 10 min at 4 °C. The clear lysate was separated into a fresh tube and centrifuged again at high speed to remove additional fat contamination.

**Immunohistochemistry**. The FUS/TLS immunohistochemistry analyses was conducted using formalin- fixed, paraffin-embedded tissue sections with an automated immunostaining platform. The assay was developed on the Ventana Discovery XT platform (Ventana Medical Systems, Inc.). The primary rabbit polyclonal FUS/TLS antibody (Cat# 11570–1-AP), purchased from Proteintech was used at a 1:50 dilution for 1 h at room temperature. On the Discovery XT platform, heat-induced antigen retrieval was conducted using the CC1 standard program and a pH9, Tris-based buffer (VMSI). Primary antibody was detected using the Discovery ChromoMap DAB (diaminobenzidine) Kit (VMSI) and Discovery OmniMap anti-rabbit HRP (VMSI). The anti-rabbit horseradish peroxidase secondary antibody was applied for 12 min at room temperature. Slides were counterstained with hematoxylin (VMSI) for 12 min at 37 °C. Hematoxylin was enhanced with bluing reagent (VMSI) for 4 min at room temperature.

**Motor neuron differentiation**. Motor neurons were differentiated from iPSCs and H9-hESCs (WiCell Research Institute and VIB-KU Leuven[35]), according to established methods with some modifications[35]. Briefly, iPSC clones were suspended and transferred from a 60-cm dish into a T-25 flask with neuronal basic medium (mixture of 50% Neurobasal medium and 50% DMEM/F12 medium, with N2 and B27 supplements without vitamin A), following collagenase type IV digestion. After 2 days incubating with 5 μM ROCK Inhibitor (Y-27632, RI, from Merck Millipore), 40 μM TGF- β inhibitor (SB 431524, SB, Tocris Bioscience), 0.2 μM bone morphogenetic protein inhibitor (LDN-193189, LDN, from Stemgent), and 3 μM GSK-3 inhibitor (CHIR99021, CHIR, from Tocris Bioscience), suspended cell spheres were then incubated with a neuronal basic medium containing 0.1 μM retinoic acid (RA, from Sigma) and 500 nM Smoothened Agonist (SAG, from Merck Millipore) for 4 days. Cells were then incubated for 2 days in a neuronal basic medium containing RA, SAG, 10 ng/ml Brain-derived neurotrophic factor (BDNF, from Peprotech), and 10 ng/ml Glial cell-derived neurotrophic factor (GDNF, from Peprotech). Cell spheres were then dissociated with a neuronal basic medium containing trypsin (0.025%)/DNase in water bath for 20 min at 37 °C, and then pipetted into single cells with the medium containing trypsin inhibitor (1.2 mg/ml). After cell counting, a defined number of cells were seeded into 20 μg/ml Laminin (Life technologies) -coated dishes or chamber slides and incubated for 5 days in a neuronal basic medium containing RA, SAG, BDNF, GDNF, and 10 μM DAPT, then incubated for 2 days in a neuronal basic medium containing BDNF, GDNF, and 20 μM Inhibitor of γ-secretase (DAPT, from Tocris Bioscience). For motor neuron maturation, cells were then kept for over 7 days in a medium containing BDNF, GDNF, 10 ng/ml ciliary neurotrophic factor (CNTF, from Peprotech).

**MTT and clonogenic survival assay**. SH-SY5Y or HEK293 cells were transfected twice with control or FUS shRNA with an interval of 24 h between transfections. 24 h after the second transfection, transfectants were trypsinized and plated in 96-well plates. Following GO or mock treatment, cell viability was measured with a TACS MTT Cell Proliferation Assay kit (Trevigen), according to the manufacturer's instructions. Cell proliferation was measured using a microplate reader (BioRad Model 680) at an absorbance of 570 nm. For the clonogenic survival assay, GO treatment was performed at various concentrations, then defined numbers of control or FUS shRNA transfected cells were plated in 6-well plates. After 10–14 days, cells were stained for 30 min with crystal violet (0.1%) and survival fractions were calculated[56].

**Single cell gel electrophoresis (Comet) Assay**. Alkaline or neutral comet assays were performed using Comet Assay kits (Trevigen), according to the manufacturer's instructions. The nuclear DNA was stained with SYBR green dye for 10 min and visualized with a fluorescent microscope (Zeiss Axio observer). Tail moment was measured by CASP software.

**Transfection, immunoblotting, co-immunoprecipitation, and immunofluorescence**. Plasmids were transfected into SH-SY5Y or HEK293 cells with Lipofectamine 2000 (Invitrogen), per the manufacturers' instructions. Immunoblotting, co-immunoprecipitation, and immunofluorescence were performed as regularly[56,57]. For immunoblotting, cell lysates extracted with lysis buffer (Fisher) containing the protease inhibitor cocktail (Roche) were loaded into 4–12% Bis-Tris precast gels (Bio-Rad) for electrophoresis. Following transferring onto the nitrocellulose membrane and incubating with primary and secondary antibodies, protein signal was detected by adding chemiluminescence reagents (Thermo) and visualized by X-ray film in dark room. Co-immunoprecipitation was performed using 2 μg of antibodies for 1 mg of total cell lysate and protein G sepharose (Sigma-Aldrich) was used to pull-down immunocomplexes followed by washing by NP-40 buffer. For immunofluorescence, cells grown on chamber slides were fixed with 4% paraformaldehyde for 15 min, followed by permeabilization in 0.5% Triton X-100 for 10 min. After incubating with primary antibodies overnight and fluorescent labeled secondary antibodies for 2 h, immunofluorescence images were captured by Zeiss Axio observer fluorescent microscope.

**In situ proximity ligation assay**. A Duolink PLA kit (Sigma) was used for the in situ PLA assay, following the manufacturer's instructions[37,58]. Briefly, cells grown in chamber slides were fixed with 3.5% formaldehyde for 15 min at 37 °C, permeabilized with 0.5% TritonX-100 for 10 min, and then incubated with primary antibodies overnight. Subsequently, cells were incubated at 37 °C with PLA probes for 1 h, with ligase for 30 min, and with polymerase for 100 min. Slides were mounted with Mounting Medium containing DAPI and PLA signal was visualized with a fluorescent microscope (Zeiss Axio observer). The negative control was tested by incubating with IgG.

**Long amplicon PCR and PCR products quantitation**. Genomic DNA was isolated using Qiagen Blood and Tissue kit per manufacturer´s directions for long amplicon PCR was performed[59]. In this study, the *hPRT* gene fragment (10.kb of encompassing exons 2–5, accession number J00205) was amplified by LongAmp Taq DNA polymerase (New England Biolabs) using forward primers 5′-TGGGGATTA CACGTGTGAACCAACC-3′ and reverse primers 5′-GCTCTACCCTGTCCTCT ACCGTCC-3′[60]. As a control, a short fragment of 250 bp of the *hPRT* gene was amplified using forward primer 5′-TGCTCGAGATGTGATGAAGG-3′ and reverse primer 5′-CTGCATTGTTTTGCCAGTGT-3′[61]. PCR products were separated in agarose gel and visualized by Gel Logic 2200 imagining system (Kodak). PicoGreen dsDNA Quantitation Kit (Molecular Probes) was used to quantify the PCR products. After PCR cycles are finished, 10 μl of the PCR products were loaded into 96-well plate followed by adding 90 μl of 1× TE buffer, which was subsequently mixed with 100 μl of the diluted PicoGreen reagent. Keep the plate in dark for 10 min at room temperature, and the fluorescence emission was read by TECAN infinite M1000 microplate reader.

**Protein purification and GST pull-down**. GST-FUS and GST-fused FUS domain polypeptides were expressed in the bacterial cell strain BL21 strain (Merck Millipore) and purified using glutathione-Sepharose 4B beads (GE Healthcare)[62]. XRCC1/LigIII complex and XRCC1 were purified from insect cells after the infection with XRCC1 and LigIII baculoviruses[63,64]. LigIII was overexpressed in and purified from bacteria as described[65]. PARP-1 was expressed as His-tag protein and purified using Ni-agarose column, followed by a cation exchange[66]. For GST pull-down, the GST alone or GST-FUS purified protein was incubated with PARP-1, XRCC1/LigIII complex or individual XRCC1 and LigIII at 4 °C overnight with EZview Red Glutathione Affinity Gel (Sigma), and beads were then washed for three time with TEN buffer (20 mM Tris·HCl, pH 7.4, 0.1 mM EDTA and 100 mM NaCl) and boiled in 4 × SDS loading buffer, and the proteins were analyzed by immunoblotting with the indicated antibodies[62].

**In vitro ligation activity assay and kinetic activity analysis**. For the ligation activity assay, DNA oligos (p24-Cy3-GGCACGGTCTACACGGCACACGAG, p27-TGTACATGATACGATTCCAAGCTAAGC, and p51-CCGTGCCAGATG TGCCGTGTGCTCACATGTACTATGCTAAGGTTCGATTCG) were synthesized by Sigma. The in vitro ligation activity assay followed three steps: (1) annealing of oligomers, (2) ligating of the nick, and (3) detection of ligation. In step1, 10 pmol of each oligomer was incubated with 50 mM NaCl in boiling water, until the water cooled down to the room temperature. In step 2, annealed oligomers were mixed with various purified proteins in 1× T4 ligation buffer and the mixture was incubated in a water bath for 20 min at 30 °C. In step 3, samples were mixed with 2× TBE sample buffer, heated for 3 min at 100 °C, and cooled down on ice for 3 min. Oligomers were then separated by denaturing urea polyacrylamide gel electrophoresis. The band with Cy3 fluorescence was detected by Typhoon FLA 7000 Ligation Kinetic analysis[38,62].

**Antisense oligonucleotide-mediated FUS knockdown**. Scrambled ASO and ASO for FUS were purchased from Exiqon (Vedbaek, Denmark), and the delivery into motor neurons was performed by adding sterile water dissolved-ASOs into culture medium of motor neuronal cells from the 20th days of differentiation from iPSC and consist maintaining ASOs for one week[35].

**Generation of CRISPR/Cas9-based FUS knockout cell line**. Single-guide RNA (sgRNA) against the *FUS* gene was designed by screening the target sequence with the online tool http://www.broadinstitute.org/rnai/public/analysis-tools/sgrna-design. One high-score sgRNA target sequence was detected in exon 4 which targeted the sense strand sequence 5′-GGAACTCAGTCAACTCCCCA-3′ towards the 5′-end of *FUS* CDS. The sgRNA module was generated by overlapping PCR protocol with minor modifications[67], and was subsequently cloned into the same pLX-sgRNA vector. Transfection of humanized Cas9 that contained lentiviral pCW-Cas9 and customized pLX-sgRNA plasmids into HEK293 cells, and cells were selected by Zeocin (505 μg/ml) and Blusticidin (5 μg/ml)[67].

Primers used to amplify U6 promotor target sgRNA sequence and terminator sequence:

Outer primer F1: AAACTCGAGTGTACAAAAAAGCAGGCTTTAAAG
Outer primer R2: AAAGCTAGCTAATGCCAACTTTGTACAAGAAAGCTG
sgRNA1Fus_R1: TGGGGAGTTGACTGAGTTCCGGTGTTTCGTCCTTTCC
sgRNA1Fus_F2: GGAACTCAGTCAACTCCCCAGTTTTAGAGCTAGAAAT AGCAA
Sequencing primers for pLX-sgRNA:
Forward: CGGGTTTATTACAGGGACAGCAG
Reverse: TACCAGTCAATCTTTCACAAATTTTGT

**Generation of isogenic controls of iPSC**. ALS patient-derived iPSCs carrying R521H mutation or P525L mutation in FUS were corrected by CellSystems (Troisdorf, Germany). iPSCs were transfected with gRNA vector, Cas9 vector, and donor DNA. Transfected cells were selected with puromycin for 4 days. Single clones were genotyped with genomic DNA PCR and subsequently sequenced. The absence of the FUS mutation was confirmed by sequencing.

**In situ apoptosis detection**. TUNEL staining was used to analyze apoptosis in motor neurons. TUNEL Assay Kit-In situ BrdU-Red DNA Fragmentation (Abcam) NeuroTACS II in situ Apoptosis Detection kit (TREVIGEN) was used for the staining. Apoptotic motor neurons seeded on chamber slides were detected following H2O2 treatment, according to the manufacturer's instructions. Images were taken by fluorescent microscope (Zeiss Axio observer).

**ADP-ribosylation assay in vitro**. The ADP-ribosylation assay was performed as per published protocol[44] with modification. Briefly, 1 μg human PARP1 purified protein was incubated with 1 μM $NAD^+$ (Sigma) and 5 pM activator oligonucleotide (5′-GGAATTCC-3′) in reaction buffer for 15 min at 37 °C, in a total reaction volume of 25 μl. Samples were then loaded in NuPAGE 4–12% Bis-Tris Protein Gels (Invitrogen) following boiling with NuPAGE LDS loading buffer (Invitrogen) for 5 min at 95 °C. The ADP-ribosylation was detected by western blotting by probing anti-Poly (ADP-Ribose) Polymer antibody (abcam).

**Statistical analysis**. A minimum of three independent experiments based on three different differentiation batches was always performed. Statistical analysis was performed using Microsoft excel or graphpad prism software. Results were analyzed for significant differences using ANOVA procedures and Student's *t*-tests, with $p < 0.05$ considered statistically significant (*$p < 0.05$, ** $p < 0.01$, *** $p < 0.001$).

## Data availability

The data that support the findings of this study are available from the corresponding author upon reasonable request.

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

## Acknowledgements

This research was primarily supported by NINDS-NIH (R01 NS088645), Muscular Dystrophy Association (MDA 294842), ALS Association, and Houston Methodist Research Institute grants to M.L.H. W.G, T.V., and L.V.D.B. received support for this work from the KU Leuven ('Opening the Future' and C14/17/107), the "Fund for Scientific Research Flanders" (FWO-Vlaanderen), the Agency for Innovation by Science and Technology (IWT; SBO-iPSCAF) and the ALS Liga (Belgium). A.E.T is supported by NIH grants NIH grants ES012512 and CA92584. We thank Dr. X. Xu for providing the Microscopy Core at the Beijing Key Laboratories for microirradiation/laser ablation experiments. The authors thank other current and former members of Hegde laboratory, specifically, Ivan de la Riva, MD/PhD summer trainee from Monterey Tech, Mexico and Priyadarshini Basu for various help. Control and sporadic ALS spinal cord tissue specimens were provided by the Department of Veterans Affairs Biorepository (VA Merit Review BX002466), South Arizona State University. Expert input from Dr. Stanley Appel, Houston Methodist Neurological Institute is greatly acknowledged.

## Author contributions

H.W. designed and performed the majority of experiments and co-wrote the manuscript. M.L.H. designed and supervised the study, analyzed and interpreted the data, and co-wrote and prepared the final manuscript. J.M. generated the CRISPR/Cas FUS KO cell line. P.M.H. contributed to purification of recombinant proteins. B.E.E. assisted in XRCC1 and LigIII knockdown assays. S.M. provided expert inputs on oxidative genome repair and commented on the manuscript. W.G., T.V., and L.V.D.B. contributed to the ALS patient-derived fibroblasts, iPSCs and the motor neuron differentiation protocol and commented on the manuscript. A.E.T. provided XRCC1/LigIII protein complex purified from insect cells, provided expert input on DNA ligation assays, and commented on the manuscript. All authors discussed the results and provided comments.

## Additional information

**Competing interests:** The authors declare no competing interests.

