## [Peer Review File · Nature Communications]

Reviewers' Comments:

Reviewer #1:

Remarks to the Author:

This manuscript from Wang et al. describes a range of experiments focused on understanding a role of the FUS hnRNP factor in a DNA ligation deficiency associated with the human disease ALS. The manuscript describes an emphasis on the DNA ligase III and XRCC1 complex and the use of various molecular and cell biology experiments to attribute functions to FUS in the ligation step of repair of oxidatively generated DNA lesions. The results follow logically along the theme that FUS is able to stimulate the DNA ligation activity of DNA ligase III both in vitro and in vivo. Supporting results with samples from patients and with ALS associated variants in the FUS protein are also described. Overall, the results described are interesting and will be useful to researchers in the fields of ALS research and DNA repair/genome stability research.

The manuscript is well written and easy to follow, although the model in Figure 10 adds nothing to the story and could be deleted.

Reviewer #2:

Remarks to the Author:

The manuscript by Wang et al. describes comprehensive and elegant set of experiments that unequivocally demonstrated the importance of FUS for efficient function of SSBR system and that FUS function is critical for high efficiency of SSBs rather than DSBs repair. The role of FUS in the SSB repair mechanism has not been previously addressed properly despite its obvious importance for neuronal homeostasis. Thus, obtained results are original and shed new light on molecular mechanisms of pathology in FUS-ALS and potentially of certain other types of ALS. Results of this study will be of high interest not only to researchers working in the field of neurodegeneration but also to wider audience and therefore deserve to be published. However, there are two major and several minor points that should be addressed before the manuscript can be accepted for publication.

Major points:

1. FUS pathology in patients used in the study should be better characterised and demonstrated. Cases used in this study were not FUS-ALS cases - in the Discussion authors state that they are sporadic - but have FUS inclusions been found in motor neurons of these patients? Most importantly, why the observed change of electrophoretic mobility was interpreted as FUS oligomerisation and a sign of FUS pathology? This is not convincing at all and overall this part of the study is quite weak. I would suggest to remove this section from the manuscript completely as it creates uncertainty and does not add much to the main idea of the study. If

authors really want to demonstrate a link between FUS pathology and SSBR system in ALS patient spinal cord neurons, they should use samples with obvious FUS inclusions and some degree of nuclear FUS clearance, and histology-based, direct or indirect methods of SSBR activity detection to demonstrate that both events take place in the same cell.

2. Authors believe that “the ligation defect in mutant FUS-P525L cells can be attributed to the increased nuclear clearance of the mutant version” but this statement is not sufficiently supported by experimental data. Moreover, as both R521H and P525L mutant proteins are not able to function efficiently in SSBR, their presence in the nucleus should not be critical. It seems that more important is haploinsufficiency that takes place in cells with a heterozygous mutation and makes mutant FUS inactive or much less active than WT protein in SSBR. If authors possess cells heterozygous for KO allele, it should be directly compared with cells carrying R521H or P525L mutations, in the ideal situation isogenic cells with KO of only the mutant allele should be used, but other option can also be employed. If the repair process in heterozygous cells will be more efficient than in mutant cells, one might consider an option of a dominant-negative effect of mutant FUS variants on WT protein function.

Minor points:

After showing nice iPSC-derived motor neurons in Figure 2d, there is no need to have Figure 6f – this panel should be moved to Supplementary Materials.

In Figure 6h, I “0” point quite obviously represents control, untreated cells and not cells 0 min after release from exposure to GO - this should be made clearer in the Figure legend.

Images in Figure 6j are of poor quality and I am not convinced that what are marked by arrows in the right column images are indeed TUNEL signals and not just debris.

Has correction of FUS mutations in mutant FUS iPSC lines reverted the delayed repair of DNA strand breaks in iPSC-derived neurons? Although not compulsory, these data would nicely complement ligation assay data.

Reviewer #3:

Remarks to the Author:

The body of work submitted by Wang et al, “Mutant FUS causes DNA ligation defects to inhibit repair of oxidative genome damage in Amyotrophic Lateral Sclerosis” put together makes an argument for the intimate involvement of FUS with activating the Xrcc1-DNA Lig3 complex to mediating DNA ligation during DNA

single-strand break repair of oxidative breaks. However, the only original major finding in this body of work is that FUS activates DNA Lig3 and that 2 common FUS mutations have independent mechanisms by which Lig3 is rendered non-functional (altered FUS-Lig3 protein interactions and FUS segregation to the cytoplasm). The authors' argue, based on their data, that FUS-involved ALS may partially involve failure in DNA Lig3 and DNA single-strand break repair.

A deeper analysis of the published data in this field from the last 4 yrs, particular PMID: 24049082, 24509083, 28082870 and 23833192 reveals that much of the data shown in the present study is confirmation of these previous reports with some incremental extension. While the authors mention that FUS has been primarily implicated in repair of DNA double-stranded breaks, there is already a great deal of data demonstrating it's involvement in oxidative breaks, particularly UVA and IR studies (which is a form of oxidative damage in of itself). Therefore, the authors' data with GO is simply an extension of this previous data using another nicking agent. Furthermore, interactions between FUS-PARP and FUS-XRCC1 have already been previously identified with some limited characterization/mapping. The authors fail to fully delineate the nature of these interactions, particularly the FUS-XRCC1 and FUS-Lig3 interactions, ie. Is PARP1 actually PARylating FUS in facilitating XRCC1/Lig3-mediated accumulation at the breaksite?. FUS has previous been shown to interact with PAR; is PARylated Xrcc1 interacting with FUS (or PARylated FUS)? Are the FUS-X1/L3 molecular interactions maintained in the presence of PARP inhibitor? Previous data from the Caldecott lab has also shown the need for PARP activity in FUS localization to DNA breaksites, again reducing the novelty of the authors' findings.

Furthermore, there are some limitations to the interpretation of the data presented by the authors. Much of these are due to questionable assumptions (DNA Lig1 is involved in BER in non-cycling cells and can in fact act redundantly with DNA Lig3) and due to a lack of appropriate controls (importantly, a primary control lacking through much of these studies was the use of Lig3 knockdown/knockout). Since the authors propose that loss of FUS function/FUS dysfunction results in a failure of Lig3-mediated ligation, a highly appropriate negative control for comparison for much of the biochemical and cellular studies would be Lig3-deficiency.

One confounding point is that ALS is predominantly a neurological deficit. Neuroblastoma and HEK293 cells do not seem like appropriate cell lines for these studies. Notwithstanding this point, the authors conclude an essential need for FUS in mediating XRCC1-LIG3 activation and localization to DNA breaksites and efficient repair. Do FUS knockdowns/knockouts in HEK293 cells, FUS/ALS patient fibroblasts, un-differentiated iPSC cells and differentiated neurons show DNA repair deficits and cell viability defects like the neuroblastoma knockdown data? If FUS dysfunction results in DNA repair defects generally, what accounts for the neurological defects? Are defects attenuated in non-neural cells and likewise enhanced in high oxygen

utilizing cell types like neurons?

In figure 3a, although the Xrcc1 IB following GST-FUS pull down seems appropriate, how do the authors reconcile that the apparent molecular weight of Lig3 in their blots (~90 kDa) is far below the known/published size (~110kDa) in the literature and from within the three anti-Lig3 abs available from Abcam. In this regard, it would have been useful for the authors to indicate molecular weight of all IB bands in all immunoblots (ie. figs 1, 2 and 3) and provide the full sized uncropped immunoblots as supplemental data. Finally, why do the authors not show the original mass spec data that identified the protein complexes?

Other points:

The authors indicate via Figs 1a/b that FUS KD cells accumulate significant DNA damage. What are the relative steady state levels of gamma-H2AX levels in the same cells that underwent comet analysis? With this level of damage, are the FUS KD cells proliferating or showing augmented cell death? While clonogenics show "relative" survival following GO, it is important for context to show steady state (untreated) proliferation and cell death levels in FUS KD vs control.

In Fig 2a, the authors should show the input/expression of FLAG-FUS transfection using anti-flag/FUS abs as an appropriate control. This will indicate the relative level of FUS overexpression compared to endogenous FUS. Furthermore, do the authors find a FUS-PARP1 interaction in the transfected cells in addition to the endogenously-expressed/IPed cells? More information as to the nature of the FUS interaction with PARP1/X1/L3 is required? Are these direct interactions? Why are the interaction domains not mapped?

Is FUS involved in Lig3-dependent mitochondrial DNA maintenance? An important corollary is whether X1 is required for the Lig3-FUS interaction and X1 is not present in the mitochondria, therefore; Lig3-FUS IPs from mitochondrial extracts along with ligation assays from purified FUS-deficient mitochondria would provide this needed insight.

In figure 2g, the X1/Lig3 PLA combination is an important control both from a methodological standpoint and as an indicator of relative X1-L3 complex levels compared to FUS-containing complexes. Furthermore, knockdown of XRCC1 and LIG3 should be used to confirm specificity of the antibodies used for PLA.

Minor point:

p.2 - Notably, mutation at R521 shows only moderate nuclear clearance, but the P525 mutation shows more robust nuclear clearance.

Manuscript# NCOMMS-17-30346

The manuscript was reviewed by two reviewers (reviewer-2 and reviewer-3)

Our point-by-point responses to the Reviewers' Comments:

New data included in the revised manuscript

Figure 2a

Figure 3a

Figure 5c

Figure 6i-j,

Figure 7b, e

Figure 8f-j

Supplementary Figure S1a-b

Supplementary Figure S2c-e

Supplementary Figure S3a-b

Supplementary Figure S5a-b

Supplementary Figure S7d

Supplementary Figure S8d-g

Supplementary Figure S9a-d

Supplementary table 3

Reviewer#2

General Comment: The manuscript by Wang et al. describes comprehensive and elegant set of experiments that unequivocally demonstrated the importance of FUS for efficient function of SSBR system and that FUS function is critical for high efficiency of SSBs rather than DSBs repair. The role of FUS in the SSB repair mechanism has not been previously addressed properly despite its obvious importance for neuronal homeostasis. Thus, obtained results are original and shed new light on molecular mechanisms of pathology in FUS-ALS and potentially of certain other types of ALS. Results of this study will be of high interest not only to researchers working in the field of neurodegeneration but also to wider audience and therefore deserve to be published. However, there are two major and several minor points that should be addressed before the manuscript can be accepted for publication.

Response: We thank the Reviewer for the appreciative comments and recognizing the novelty of these findings, and for the favorable recommendation for publication. We also thank the Reviewer for suggesting two major points to be addressed, including IHC data of human spinal cord tissue and haploinsufficiency versus dominant negative nature of mutant FUS. We have included new data on these in the revised manuscript as described below in our point-by-point responses to the specific comments. These revisions have significantly improved the manuscript.

Major Comment-1: FUS pathology in patients used in the study should be better characterized and demonstrated. Cases used in this study were not FUS-ALS cases - in the Discussion authors state that they are sporadic – but have FUS inclusions been found in motor neurons of these patients? Most importantly, why the observed change of electrophoretic mobility was interpreted as FUS oligomerisation and a sign of FUS pathology? This is not convincing at all and overall this part of the study is quite weak. I would suggest to remove this section from the manuscript completely as it creates uncertainty and does not add much to the main idea of the study. If authors really want to demonstrate a link between FUS pathology and SSBR system in ALS patient spinal cord neurons, they should use samples with obvious FUS inclusions and some degree of nuclear FUS clearance, and histology-based, direct or indirect methods of SSBR activity detection to demonstrate that both events take place in the same cell.

Response: While we agree with the Reviewer that the data from human ALS spinal cord tissue may not show a direct connection between the observed genome damage and FUS pathology, we believe that these data particularly with the substantially reduced DNA ligation activity (Figure 5e-f) and reduced genomic DNA integrity (Figure 5d) observed in ALS spinal cord tissue provide a strong correlation between nuclear clearance of FUS and accumulation of genome damage. Moreover, based on the Reviewer's suggestion, we have now performed immunohistochemistry (IHC) of control and ALS spinal cord tissue sections using FUS antibody (Figure 5c). The ALS spinal cord showed clear and significant cytosolic accumulation of FUS, similar to the FUS pathology previously demonstrated in ALS-FUS patients^{1,2}. With respect to the immunoblots of ALS spinal cord tissue (Figure 5a), these show a reduction in monomeric FUS along with an increase in high mobility bands (oligomeric FUS). This has been clarified in the revised text.

With the new IHC data clearly showing FUS pathology, along with data showing DNA repair defects in human ALS-FUS spinal cord, we believe that we have provided important *in vivo* support for our model. We have therefore retained Figure 5 as part of the main figures of the revised manuscript. However, we are willing to move Figure 5 to Supplementary Material based on the Reviewer's opinion.

Major Comment-2: Authors believe that “the ligation defect in mutant FUS-P525L cells can be attributed to the increased nuclear clearance of the mutant version” but this statement is not sufficiently supported by experimental data. Moreover, as both R521H and P525L mutant proteins are not able to function efficiently in SSBR, their presence in the nucleus should not be critical. It seems that more important is haploinsufficiency that takes place in cells with a heterozygous mutation and makes mutant FUS inactive or much less active than WT protein in SSBR. If authors possess cells heterozygous for KO allele, it should be directly compared with cells carrying R521H or P525L mutations, in the ideal situation isogenic cells with KO of only the mutant allele should be used, but other option can also be employed. If the repair process in heterozygous cells will be more efficient than in mutant cells, one might consider an option of a dominant-negative effect of mutant FUS variants on WT protein function.

Response: This is an important suggestion. We greatly appreciate the Reviewer for pointing this out, as determining the haploinsufficiency versus dominant negative effect of the FUS mutation will be critical for future intervention strategies. However, we do not possess cells containing allele-specific FUS knockout (KO), and anticipate that there will be technical difficulties with this approach, due to the strong autoregulatory property of FUS, that are similar to TDP-43 (reviewed extensively in our recent publication³).

Therefore, to address this important issue, we designed two independent approaches. (1) First, we optimized ~50% transient knockdown (KD) of FUS in wildtype human motor neurons using antisense oligonucleotides (ASOs) and measured DNA Ligase III (LigIII) activity in control versus 50% FUS KD cells. 50% KD was confirmed by both mRNA quantitation (Figure 8f) and immunoblotting to assess protein level (Figure 8g). Incubation with 15nM ASOs for approximately one week caused ~50% KD of FUS in motor neurons. The 50% FUS KD cells showed a moderate (~20%) reduction in LigIII activity (Figure 8i and 8j; Lane 1 vs. 2). We then compared LigIII activity in the patient derived R521H mutant (heterozygous) FUS cell line vs. the FUS KD line. The heterozygous line derived from an ALS patient is expected to have 50% wildtype and 50% mutant FUS. However, the ligase activity in mutant motor neurons was significantly reduced compared to both the wildtype and 50% KD lines. Quantitation of ligase activity from three independent experiments showed that the LigIII activity was ~50% lower in the mutant cells than in wildtype cells, and a further ~30% reduced compared to FUS KD cells (Figure 8i). These data indicated a dominant negative effect of mutant FUS on LigIII activity, rather than only haploinsufficiency. For haploinsufficiency effect alone, one would expect LigIII activity to be comparable to that of the 50% KD cell line. (2) In a complementary approach, we used controlled ectopic expression of wildtype and mutant FUS at a comparable level (~2-fold), and measured LigIII activity in XRCC1 IP complex (Supplementary Figure S8d-g). We generated inducible cell lines, stably transfected with wildtype and R521H mutant FUS. The cells were induced with doxycycline (2µg/µl) for one week. Here, wildtype FUS overexpression increased LigIII activity by ~20%, whereas, mutant (R521H) expression decreased ligase activity by about 25% compared to control cells. Taken together, these data demonstrate a dominant negative effect of mutant FUS on LigIII activity.

The dominant nature of toxic gain of function of FUS mutants indicated by our studies is consistent with recent *in vivo* FUS KO and mutant transgenic mice studies. Although, a FUS KO mice was viable and did not develop strong ALS-like phenotype⁴, FUS mutant transgene expression in mice induced selective motor neuron degeneration⁵.

We have included these new data and related text in the revised manuscript.

Minor comments:

Minor comment-1: After showing nice iPSC-derived motor neurons in Figure 2d, there is no need to have Figure 6f – this panel should be moved to Supplementary Materials.

Response: We have now moved Figure 6f to Supplementary Material (Supplementary Figure S6a).

Minor Comment-2: In Figure 6h, I "0" point quite obviously represents control, untreated cells and not cells 0 min after release from exposure to GO - this should be made clearer in the Figure legend.

Response: This has now been corrected in the revised manuscript.

Minor Comment-3: Images in Figure 6j are of poor quality and I am not convinced that what are marked by arrows in the right column images are indeed TUNEL signals and not just debris.

Response: We apologize for the poor quality of the TUNEL images. We have now repeated the TUNEL experiment using a different TUNEL assay kit (purchased from abcam. Cat.# ab66110), which allows fluorescence detection of TUNEL signals, and thereby yielding better quality images. The new data (Figure 6i-j) clearly shows significantly increased TUNEL signals in FUS-mutant expressing cells.

Minor Comment-4: Has correction of FUS mutations in mutant FUS iPSC lines reverted the delayed repair of DNA strand breaks in iPSC-derived neurons? Although not compulsory, these data would nicely complement ligation assay data.

Response: We again thank the Reviewer for the suggested experiment. We performed LA-PCR based DNA integrity measurements at early (30min) and late (180min) time points after GO treatment. The data at 30min show a comparable level of DNA damage (40-50%) induced by GO. At 180 min, DNA integrity was mostly restored in mutation-corrected cells (~90% DNA integrity). However, mutant cells still showed significantly reduced (60%) DNA integrity. These data included in the revised manuscript (Figure 7e) confirm that the observed ligation defect and delayed repair are indeed caused by FUS mutations.

Reviewer #3

General Comment: The body of work submitted by Wang et al, "Mutant FUS causes DNA ligation defects to inhibit repair of oxidative genome damage in Amyotrophic Lateral Sclerosis" put together makes an argument for the intimate involvement of FUS with activating the Xrcc1-DNA Lig3 complex to mediating DNA ligation during DNA single-strand break repair of oxidative breaks. However, the only original major finding in this body of work is that FUS activates DNA Lig3 and that 2 common FUS mutations have independent mechanisms by which Lig3 is rendered non-functional (altered FUS-Lig3 protein interactions and FUS segregation to the cytoplasm). The authors' argue, based on their data, that FUS-involved ALS may partially involve failure in DNA Lig3 and DNA single-strand break repair. A deeper analysis of the published data in this field from the last 4 yrs, particular PMID: 24049082, 24509083, 28082870 and 23833192 reveals that much of the data shown in the

present study is confirmation of these previous reports with some incremental extension. While the authors mention that FUS has been primarily implicated in repair of DNA double-stranded breaks, there is already a great deal of data demonstrating its involvement in oxidative breaks, particularly UVA and IR studies (which is a form of oxidative damage in of itself). Therefore, the authors' data with GO is simply an extension of this previous data using another nicking agent. Furthermore, interactions between FUS-PARP and FUS-XRCC1 have already been previously identified with some limited characterization/mapping. The authors fail to fully delineate the nature of these interactions, particularly the FUS-XRCC1 and FUS-Lig3 interactions, ie. Is PARP1 actually PARylating FUS in facilitating XRCC1/Lig3-mediated accumulation at the breaksite?. FUS has previous been shown to interact with PAR; is PARylated Xrcc1 interacting with FUS (or PARylated FUS)? Are the FUS-X1/L3 molecular interactions maintained in the presence of PARP inhibitor? Previous data from the Caldecott lab has also shown the need for PARP activity in FUS localization to DNA break sites, again reducing the novelty of the authors' findings.

Response: We appreciate and thank the Reviewer for the critical but constructive issues raised. Although recent studies have suggested the involvement of FUS in genome maintenance and the DNA damage response (DDR), these studies were limited to analysis of DNA damage accumulation and deficient double-strand break repair (DSBR) in FUS inhibited/mutated cells. The precise mechanism of FUS in the DDR, as well as the linkage between ALS-FUS mutations and motor neuronal cell death due to DNA repair deficiency has not been demonstrated. Although PARP-1 dependent recruitment of FUS through interaction with PAR has been shown recently, mostly in tumor-derived cell lines^{6,7}, its relationship to specific DNA repair pathway or in downstream repair reactions were not investigated. Our studies provide important molecular insights into the specific inhibition of XRCC1/LigIII recruitment at oxidative genome damage sites in mutant FUS expressing human neurons or FUS KD cells, which causes defective DNA single-strand break ligation, a rate-limiting reaction in the repair of oxidative genome damage. In addition, we are not aware of any published studies describing the FUS-XRCC1 interaction, contrary to what is stated by the Reviewer. To our knowledge, ours is the first study documenting both the physical and functional association of FUS with XRCC1/LigIII complex. Furthermore, although previous studies have suggested a role for FUS in the repair of DSBs, the relative presence of DSB versus non-DSB damage caused by loss of FUS or FUS mutations was not known before. One study⁶ showed co-localization of FUS with γ -H2AX at laser (405nm) ablation sites to suggest its possible presence at DSB sites. Another report⁷ showed its presence at UVA (351nm) damage sites to suggest linkage to oxidative damage, even though the DSB marker γ -H2AX was again used to detect DNA damage. It is important to note that laser ablation does not specifically induce a single-type of damage. Although, laser irradiation at 405nm is reported to induce more DSBs than at 351nm where it induces more SSBs and oxidized base lesions; however, in most cases it forms both DSB and non-DSB damage is induced to the laser track at both wavelengths. Thus it is not possible to accurately attribute the co-localization data to one type of damage.

Here we have addressed this ambiguity in literature, and quantitated DSB and non-DSB damage by a combination of alkaline and neutral comet assays in unstressed FUS KD cells

and demonstrate that the majority of damage accumulated in the absence of FUS are single-strand breaks. A small number of DSBs may be generated secondarily due to the accumulation of unrepaired alkali labile sites^{8,9}. Thus our study clearly demonstrates the linkage between the loss of FUS and oxidative DNA lesions other than DSBs, in neurons. This is an important finding because the two types of damage not only utilize distinct repair pathways, but also induce distinct damage response signaling to mediate cytotoxicity/death. We thus respectfully disagree about the comments downplaying the importance and novelty of documenting of a specific DNA ligase III defect in FUS-linked ALS. To our knowledge, this is the first study identifying defects in a specific DNA repair reaction linked to ALS and FUS-associated neurodegeneration, which is critical for exploring DNA repair-targeted intervention strategies.

However, we do appreciate the Reviewer's suggestion for additional insights on this interaction including the role of PARylation. While, our *in vitro* interaction studies using purified proteins revealed a direct physical interaction between FUS and XRCC1/LigIII, which does not rule out the involvement of PAR in modulating the interaction. To address this, we performed an immunoprecipitation (IP) assay with FUS antibody from cells treated with or without PARP1 inhibitor (AG-14361) and probed for the presence of XRCC1 (Supplementary Figure S9a). It was observed that the level of XRCC1 was markedly reduced in FUS IP from AG-14361-treated cells, suggesting that the interaction is promoted by PARP-1 activity. This was also confirmed by reduced PLA signals for FUS versus PARP-1 in PARP-1 inhibitor-treated cells (Supplementary Figure S9b). It is likely that PAR on auto-PARylated PARP-1 provides the initial signal for the recruitment of FUS and XRCC1's to damage sites, and that the interaction could be stabilized by direct binding.

In addition, we also performed an *in vitro* ADP-ribosylation assay using purified PARP1 protein and incubated the reaction with NAD⁺ and octameric oligonucleotide (according to a previously published protocol¹⁰). Auto-PARylated PARP1 was detected by immunoblotting with PAR antibody (Supplementary Figure S9c). Surprisingly, when we add purified FUS protein to the reaction, the auto-PARylation level of PARP-1 was remarkably increased by ~10 folds. The PAR antibody also detected a second lower mobility band detected by PAR antibody at a size corresponding to that of FUS, which suggests that FUS may be PARylated by PARP-1 *in vitro*, consistent with a previous mass spectroscopy screening study¹¹. Nonetheless, these data suggest that FUS promotes PARP-1 activity. To test the effect of PARP-1 activity on the FUS-XRCC1 interaction, we performed *in vitro* GST affinity pulldown in the presence of PARP-1 and NAD⁺ and found that PARylation enhances the *in vitro* interaction of FUS with XRCC1 (Supplementary Figure S9d). Together, these new data suggest that PARP-1 and its PARylation activity enhance the interaction of FUS with XRCC1, which is critical for recruitment of XRCC1/LigIII to sites of oxidatively damaged DNA (Schematically shown in the revised Fig. 9g).

Major Comment-1: Furthermore, there are some limitations to the interpretation of the data presented by the authors. Much of these are due to questionable assumptions (DNA

Lig1 is involved in BER in non-cycling cells and can in fact act redundantly with DNA Lig3) and due to a lack of appropriate controls (importantly, a primary control lacking through much of these studies was the use of Lig3 knockdown/knockout). Since the authors propose that loss of FUS function/FUS dysfunction results in a failure of Lig3-mediated ligation, a highly appropriate negative control for comparison for much of the biochemical and cellular studies would be Lig3-deficiency.

Response: We have now probed FUS IP for the presence of LigI, but were unable to detect it (Supplementary Figure S5a). This suggested that the association of FUS with LigIII is specific. Moreover, LigI is primarily involved in replication-associated LP-BER^{12,13}, and thus its level is very low in postmitotic cells as with other replication-linked proteins¹⁴. Based on the Reviewer's suggestion, we also measured ligation activity in XRCC1 IP isolated from LigIII KD cells as a control; this showed a significant decrease in DNA nick ligation, as expected. Furthermore, addition of recombinant FUS did not enhance ligation activity in LigIII KD cells, confirming the specificity of LigIII activation by FUS (Supplementary Figure S5b). These data provide further support for our model of FUS-mediated regulation of LigIII activity.

Major Comment-2: One confounding point is that ALS is predominantly a neurological deficit. Neuroblastoma and HEK293 cells do not seem like appropriate cell lines for these studies. Notwithstanding this point, the authors conclude an essential need for FUS in mediating XRCC1-LIG3 activation and localization to DNA breaksites and efficient repair. Do FUS knockdowns/knockouts in HEK293 cells, FUS/ALS patient fibroblasts, undifferentiated iPSC cells and differentiated neurons show DNA repair deficits and cell viability defects like the neuroblastoma knockdown data? If FUS dysfunction results in DNA repair defects generally, what accounts for the neurological defects? Are defects attenuated in non-neural cells and likewise enhanced in high oxygen utilizing cell types like neurons?

Response: We acknowledge that the vulnerability of only the affected neurons to FUS pathology/repair defects is an important point to be investigated. In our study, we used multiple cell models, including fibroblasts and iPSCs identified from ALS-patients with FUS mutations, motor neurons differentiated from iPSCs, differentiated SH-SY5Y cells, and HEK293 cells to examine if the FUS-mediated LigIII effect is a global phenomenon across cell types or specific to neurons. The data revealed that FUS plays a key role in regulating LigIII activity in all healthy cells tested. However, the effect of loss of function of FUS because of mutation is manifested only in spinal motor neurons. Similarly, deficiencies in other DNA repair factors such as aprataxin, PNKP and ATM cause phenotypes related to specific neuronal cell types but it is not known why. Moreover, loss of nuclear LigIII function has very little effect on DNA damage sensitivity suggesting functional overlap with LigI in proliferating cells. Thus the loss of FUS may have more profound effect in post-mitotic cells than in cycling cells, which have higher LigI activity. In addition, as suggested by the Reviewer, the higher O₂ consumption, and higher transcription/metabolic activity of neurons may be critical, in addition to its cross-talk with other etiological factors linked to ALS. It is also reasonable to speculate that although the

loss of a small percent of spinal cord neurons or CNS results in motor function phenotype, other tissues may be tolerant to the loss/dysfunction of a small fraction of cells.

In any case, these important long-standing questions need to be addressed in future studies using in vivo models. We thank the Reviewer for mentioning this point, which we have now brought up in the Discussion.

Major comment-3: In figure 3a, although the Xrcc1 IB following GST-FUS pull down seems appropriate, how do the authors reconcile that the apparent molecular weight of Lig3 in their blots (~90 kDa) is far below the known/published size (~110kDa) in the literature and from within the three anti-Lig3 abs available from Abcam. In this regard, it would have been useful for the authors to indicate molecular weight of all IB bands in all immunoblots (ie. figs 1, 2 and 3) and provide the full sized uncropped immunoblots as supplemental data. Finally, why do the authors not show the original mass spec data that identified the protein complexes?

Response: The molecular weight indicated in the first panel of Figure 3a is for the Coomassie stained gel, which indicates GST-FUS, and not for the following immunoblot images that show XRCC1 and LigIII. However, we agree that our alignment of the immunoblots to this panel causes confusion about the molecular weight of LigIII. We apologize for this and thank the reviewer for pointing this out. In Figure 3a, LigIII antibody from Abcam (Cat# ab587), which recognizes LigIII at ~110kDa, as rightly pointed out by the Reviewer. We have now aligned the representation of immunoblots and indicated the molecular weight in all immunoblots as suggested by the Reviewer to avoid this confusion. Furthermore, based on the Reviewer's suggestion, we have included a table listing proteins identified in FUS IP by mass-spectrometry in Supplementary Material (Supplementary Table S3).

Minor Comment-1: The authors indicate via Figs 1a/b that FUS KD cells accumulate significant DNA damage. What are the relative steady state levels of gamma-H2AX levels in the same cells that underwent comet analysis? With this level of damage, are the FUS KD cells proliferating or showing augmented cell death? While clonogenics show "relative" survival following GO, it is important for context to show steady state (untreated) proliferation and cell death levels in FUS KD vs control.

Response: Previous studies have shown the presence of γ H2AX foci in FUS KD cells. However, the relative abundance of DSB versus non-DSB damage sites has not been addressed before. To investigate this, we analyzed the level of damage in alkaline vs neutral comet assay. At 48h after FUS shRNA transfection, the cells predominantly accumulated non-DSB damage as shown by a significant increase in alkaline comet tails but not neutral comet tails (Figure 1c). Consistent with these observations, these cells showed only a moderate increase in γ -H2AX levels (data not shown). Although in Figure 1d, 1e, and 1f, the steady state cell survival was normalized to allow for a better comparison of the sensitivity of the control and KD cells to oxidative stress, based on the Reviewer's suggestion, we examined the steady state level of cell viability and proliferation using the

MTT and clonogenic data of untreated cells and have presented these histograms in Supplementary Figure S1a-b. The MTT assay performed 72h after shRNA transfection revealed no significant change in cell viability, whereas the clonogenic assay, represented as plating efficiency showed a moderate (~5%) decrease in the average number of colonies formed. Thus these data show that in steady state, the FUS KD did not significantly affect the survival of unstressed cells, but only moderately affected cell proliferation. These data and the related text have now been included in the revised manuscript.

Minor comment-2: In Fig 2a, the authors should show the input/expression of FLAG-FUS transfection using anti-flag/FUS abs as an appropriate control. This will indicate the relative level of FUS overexpression compared to endogenous FUS. Furthermore, do the authors find a FUS-PARP1 interaction in the transfected cells in addition to the endogenously-expressed/IPed cells? More information as to the nature of the FUS interaction with PARP1/X1/L3 is required? Are these direct interactions? Why are the interaction domains not mapped?

Response: The immunoblots with anti-Flag and anti-FUS antibodies have now been included in Figure 2a. We detected PARP1 in Flag-FUS IPs, and this data are included in the revised Figure 2a.

Regarding the FUS-PARP-1 interaction, we also observed the association in FLAG-FUS expressing cells (Figure 2a) as in endogenous IP of FUS. Although we showed the direct interaction between FUS and the XRCC1/LigIII complex in the original manuscript, we have now performed broad domain mapping studies and found that aa268-aa355 in FUS is the major region involved in XRCC1 binding (Supplementary Figure S3). The C-terminal aa465-aa526 also exhibited weak binding, while the N-terminal aa1-aa267 did not show any binding. As mentioned in our response to another comment above, we also tested the effect of PARylation on these interactions. Our data show that, although FUS interacts directly with XRCC1/LigIII in the absence of PARylation, PARylation greatly enhances the interaction (Supplementary Figure S9d). Furthermore, the FUS-XRCC1 interaction was reduced in PARP-inhibitor treated cells (Figure 9a). These data have thus provided important insight on the in-cell regulation of FUS-XRCC1 interaction.

Minor Comment-3: Is FUS involved in Lig3-dependent mitochondrial DNA maintenance? An important corollary is whether X1 is required for the Lig3-FUS interaction and X1 is not present in the mitochondria, therefore; Lig3-FUS IPs from mitochondrial extracts along with ligation assays from purified FUS-deficient mitochondria would provide this needed insight.

Response: This is a natural follow up of these studies, and our initial intention was to pursue the role of FUS in mitochondrial genome maintenance in a separate and comprehensive study. As the Reviewer rightly pointed out, XRCC1 is absent from mitochondria. Mitochondrial LigIII and nuclear LigIII are generated by alternative translation initiation with the initial sequence next to the first ATG in the LigIII mRNA open reading frame encoding a MTS sequence whereas as nuclear LigIII results from

translation initiation from an internal in-frame ATG. While FUS has been shown to localize in mitochondria, although its functional role in mitochondria, particularly with respect to mitochondrial genome maintenance, has not been characterized. Following up on the Reviewer's suggestion, we performed LA-PCR on mitochondrial DNA from control and FUS KO cells. This revealed a ~25% decrease in mitochondrial DNA integrity in unstressed cells (data shown below). We next evaluated the recovery of mitochondrial DNA in control versus FUS KO cells after oxidative stress. The FUS KO cells showed a significantly reduced recovery compared to the control cells. This suggests that FUS is likely to be involved in mitochondrial DNA maintenance as well. However, our efforts to measure LigIII activity in mitochondrial extracts have not been successful, likely because of the need for a large scale up of culture required for such assays, in addition to the possible inactivation of LigIII during the multiple treatments associated with the isolation of mitochondria of high purity. These studies indicated a role for FUS in mitochondrial genome maintenance; however they require extensive optimization and scale up and therefore we believe that they should be part of a separate manuscript. However, based on the Reviewer's and Editor's discretion, we would be willing to include the data shown below in the Supplementary Material.

Figure. Loss of FUS affects mitochondrial DNA integrity and causes delayed recovery of oxidative stress-induced mitochondrial DNA damage.

- Steady state level of mitochondrial DNA strand breaks in unstressed FUS KO cells. Total DNA was isolated from control and FUS KO (CRISPR-Cas9) HEK293 cells. LA-PCR was performed for primers targeting a 8 kb mitochondrial DNA fragment¹⁵, which showed ~25% decrease in DNA integrity in after FUS KO.
- Kinetics of mitochondrial DNA recovery after oxidative stress. The control and FUS KO cells were treated with 100nM GO for 30min. Total DNA was isolated immediately and 3h after GO treatment and analyzed by LA PCR. While mitochondrial DNA integrity was recovered to ~90% in control cells at 3h, FUS cells showed only moderate recovery.

Minor Comment-4: In figure 2g, the X1/Lig3 PLA combination is an important control both from a methodological standpoint and as an indicator of relative X1-L3 complex levels compared to FUS-containing complexes. Furthermore, knockdown of XRCC1 and LIG3 should be used to confirm specificity of the antibodies used for PLA.

Response: We agree with the Reviewer and apologize for not including these controls in the original manuscript. The PLA of FUS vs XRCC1 and FUS vs LigIII in control and XRCC1 or LigIII siRNA transfected cells has now been included in Supplementary Figure 2c, 2d and 2e. These data showed decreased interaction after XRCC1 as well as LigIII KD, confirming the antibody specificity.

Minor Comment-5: p.2 - Notably, mutation at R521 shows only moderate nuclear clearance, but the P525 mutation shows more robust nuclear clearance.

Response: R521C has previously been shown to cause significant nuclear clearance of FUS¹⁶. In our study, the R521C mutation showed greater nuclear clearance compared to the R521H mutation (Figure 8c). Furthermore, P525L shows more robust nuclear clearance than R521H. No previous studies have directly compared the nuclear clearance of the R521 and P525 mutations. We have now corrected this point in the revised manuscript.

References cited in these point-by-point responses:

- 1 Kwiatkowski, T. J., Jr. *et al.* Mutations in the FUS/TLS gene on chromosome 16 cause familial amyotrophic lateral sclerosis. *Science* **323**, 1205-1208, doi:10.1126/science.1166066 (2009).
- 2 Vance, C. *et al.* Mutations in FUS, an RNA processing protein, cause familial amyotrophic lateral sclerosis type 6. *Science* **323**, 1208-1211, doi:10.1126/science.1165942 (2009).
- 3 Guerrero, E. N. *et al.* TDP-43/FUS in motor neuron disease: Complexity and challenges. *Prog Neurobiol* **145-146**, 78-97, doi:10.1016/j.pneurobio.2016.09.004 (2016).
- 4 Kino, Y. *et al.* FUS/TLS deficiency causes behavioral and pathological abnormalities distinct from amyotrophic lateral sclerosis. *Acta Neuropathol Commun* **3**, 24, doi:10.1186/s40478-015-0202-6 (2015).
- 5 Sharma, A. *et al.* ALS-associated mutant FUS induces selective motor neuron degeneration through toxic gain of function. *Nat Commun* **7**, 10465, doi:10.1038/ncomms10465 (2016).
- 6 Mastrocola, A. S., Kim, S. H., Trinh, A. T., Rodenkirch, L. A. & Tibbetts, R. S. The RNA-binding protein fused in sarcoma (FUS) functions downstream of poly(ADP-ribose) polymerase (PARP) in response to DNA damage. *J Biol Chem* **288**, 24731-24741, doi:10.1074/jbc.M113.497974 (2013).
- 7 Rulten, S. L. *et al.* PARP-1 dependent recruitment of the amyotrophic lateral sclerosis-associated protein FUS/TLS to sites of oxidative DNA damage. *Nucleic Acids Res* **42**, 307-314, doi:10.1093/nar/gkt835 (2014).
- 8 Hegde, M. L. *et al.* Studies on genomic DNA topology and stability in brain regions of Parkinson's disease. *Arch Biochem Biophys* **449**, 143-156, doi:10.1016/j.abb.2006.02.018 (2006).
- 9 Madabhushi, R., Pan, L. & Tsai, L. H. DNA damage and its links to neurodegeneration. *Neuron* **83**, 266-282, doi:10.1016/j.neuron.2014.06.034 (2014).

- 10 Schuhwerk, H. *et al.* Kinetics of poly(ADP-ribosyl)ation, but not PARP1 itself, determines the cell fate in response to DNA damage in vitro and in vivo. *Nucleic Acids Res* **45**, 11174-11192, doi:10.1093/nar/gkx717 (2017).
- 11 Zhang, Y., Wang, J., Ding, M. & Yu, Y. Site-specific characterization of the Asp- and Glu-ADP-ribosylated proteome. *Nat Methods* **10**, 981-984, doi:10.1038/nmeth.2603 (2013).
- 12 Bentley, D. J. *et al.* DNA ligase I null mouse cells show normal DNA repair activity but altered DNA replication and reduced genome stability. *J Cell Sci* **115**, 1551-1561 (2002).
- 13 Levin, D. S., McKenna, A. E., Motycka, T. A., Matsumoto, Y. & Tomkinson, A. E. Interaction between PCNA and DNA ligase I is critical for joining of Okazaki fragments and long-patch base-excision repair. *Curr Biol* **10**, 919-922 (2000).
- 14 Peng, Z., Liao, Z., Matsumoto, Y., Yang, A. & Tomkinson, A. E. Human DNA Ligase I Interacts with and Is Targeted for Degradation by the DCAF7 Specificity Factor of the Cul4-DDB1 Ubiquitin Ligase Complex. *J Biol Chem* **291**, 21893-21902, doi:10.1074/jbc.M116.746198 (2016).
- 15 Furda, A., Santos, J. H., Meyer, J. N. & Van Houten, B. Quantitative PCR-based measurement of nuclear and mitochondrial DNA damage and repair in mammalian cells. *Methods Mol Biol* **1105**, 419-437, doi:10.1007/978-1-62703-739-6_31 (2014).
- 16 Acosta, J. R. *et al.* Mutant human FUS Is ubiquitously mislocalized and generates persistent stress granules in primary cultured transgenic zebrafish cells. *PLoS One* **9**, e90572, doi:10.1371/journal.pone.0090572 (2014).

Reviewers' Comments:

Reviewer #1:

Remarks to the Author:

The authors have responded to the concerns of the reviewers in an adequate and professional fashion.

Reviewer #2:

Remarks to the Author:

I believe that authors done a good job in addressing reviewers' criticism and new experimental data substantially improved the manuscript. I am satisfied with the responses to all my comments except one, the Major point 1.

I am still not happy with the inclusion of data obtained by analysis of patients' samples. Authors' conclusion that higher molecular bands are oligomers is still not substantiated. The fact that there is a swing of band intensity from 75 kDa to the higher molecular weight bands is not a prove that the latter bands are oligomers. A number of other explanations could be suggested, e.g. ubiquitination, SUMOylation, etc.

Also, I am surprised that FUS cytoplasmic mislocalisation, though limited, was observed in sporadic ALS cases. Moreover, IHH images do not show much neuronal loss in the ventral spinal cord of ALS patients – probably histopathological assessment of these cases included estimation of the degree of motor neuron loss and this important information need to be included. And are shown regions indeed from the ventral horns? – in this aspect upper panels in Fig. 5c are useless and should be substituted by (or better, supplemented by) more general plan/lower magnification images to explicitly locate the area shown in high magnification images.

In conclusion, I still believe that data shown in Fig. 5 are week, and their removal along with corresponding text (notably, these data are barely mentioned in the rather lengthy discussion!) shall do no harm to the manuscript.

Some minor points that need to be clarified in the final version of the manuscript:

In Fig. 1g labelling of comet assay is confusing, particularly in the presence of panel c in the same Figure. The legend needs to explain why 0 min after GO results in 1g are so different from no treatment results in 1c.

The legend to Fig. 2f is misleading. Presumably the histogram shows results from several (how many?) independent IP/WB experiments because if "Quantitation of IB bands intensity in Figure 2b and 2e" was what indeed shown there where error bars came from? Secondly, I suggest to remove the word "fold" from the description of

this histogram in the legend because as presented it does not show fold change AFTER GO treatment

.

The legend to the Fig. 2h does not indicate quantification for what PLA experiments are actually shown in panel g (i.e. in SH-SY5Y cells or in motor neurons).

Reviewer #3:

Remarks to the Author:

I thank the authors in question for their thorough and thoughtful rebuttal of my critique of "Mutant FUS causes DNA ligation defects to inhibit repair of oxidative genome damage in Amyotrophic Lateral Sclerosis". I believe that the authors have done an outstanding job to address all concerns and questions. The additional data, experimental controls and discussion points have now generated a compelling study describing the critical role that FUS plays in repairing oxidative breaks and is correlative with patient samples and pathology. Furthermore, the PAR/PARPi data serves to increase the mechanistic insight of the authors' findings. I am elated to see the connection between mitochondrial DNA integrity and FUS loss; a finding that either is complementary to the present study or may in fact be more important to ALS pathology. While the authors have provided this tidbit of data, a great deal of additional work would need to be performed to extensively elucidate the mechanism of mtDNA degeneration in FUS mutants/KO and its role in the pathology of FUS-ALS. In order to not diminish the impact of this future study, I will treat the mtDNA finding as a "reviewer figure" and for the benefit of the authors, to NOT publish this additional piece data so that it may be included in their future study.

NCOMMS-17-30346A

Point-by-point response to Reviewer's suggestions

Reviewer #1

The authors have responded to the concerns of the reviewers in an adequate and professional fashion.

Response: We thank the reviewer for the appreciation and favorable recommendation.

Reviewer #2

Comment: I believe that authors done a good job in addressing reviewers' criticism and new experimental data substantially improved the manuscript. I am satisfied with the responses to all my comments except one, the Major point 1. I am still not happy with the inclusion of data obtained by analysis of patients' samples. Authors' conclusion that higher molecular bands are oligomers is still not substantiated. The fact that there is a swing of band intensity from 75 kDa to the higher molecular weight bands is not a proof that the latter bands are oligomers. A number of other explanations could be suggested, e.g. ubiquitination, SUMOylation, etc. Also, I am surprised that FUS cytoplasmic mislocalisation, though limited, was observed in sporadic ALS cases. Moreover, IHH images do not show much neuronal loss in the ventral spinal cord of ALS patients – probably histopathological assessment of these cases included estimation of the degree of motor neuron loss and this important information need to be included. And are shown regions indeed from the ventral horns? – in this aspect upper panels in Fig. 5c are useless and should be substituted by (or better, supplemented by) more general plan/lower magnification images to explicitly locate the area shown in high magnification images. In conclusion, I still believe that data shown in Fig. 5 are weak, and their removal along with corresponding text (notably, these data are barely mentioned in the rather lengthy discussion!) shall do no harm to the manuscript.

Response: We respect the Reviewer's concern regarding the precise nature of the high molecular weight bands in immunoblots of ALS spinal cords tissue extracts and attributing them to FUS oligomers, and based on the combined recommendation of the Editor and the Reviewer, we have moved Fig. 5 to Supplementary Material (new Supplementary Fig. 5). The spinal cord tissue used for IHC and immunoblotting were from cervical region and we have mentioned this in the Methods section of the revised manuscript. Furthermore, although FUS pathology is normally associated with its mutant forms in familial ALS, previous studies have shown FUS pathology in about 1-5% sporadic ALS patients. We have also included a brief statement about the significance of these data in the revised discussion.

Comment: In Fig. 1g labelling of comet assay is confusing, particularly in the presence of panel c in the same Figure. The legend needs to explain why 0 min after GO results in 1g are so different from no treatment results in 1c.

Response: We thank the reviewer for noting this. Time '0' post GO in Fig 1g is immediately after 1h of GO treatment, which shows the level of DNA damage induced by GO. The untreated control in Fig. 1c are unstressed cells and thus show only basal DNA damage. We have clarified this in the Fig. 1g as Time (min) post GO/1h and revised the figure legend accordingly.

Comment: The legend to Fig. 2f is misleading. Presumably the histogram shows results from

several (how many?) independent IP/WB experiments because if “Quantitation of IB bands intensity in Figure 2b and 2e” was what indeed shown there where error bars came from? Secondly, I suggest to remove the word “fold” from the description of this histogram in the legend because as presented it does not show fold change AFTER GO treatment.

Response: We have now mentioned that the histogram represents mean IB band intensity from three independent experiments and corrected the ‘fold’ in the figure description.

Comment: The legend to the Fig. 2h does not indicate quantification for what PLA experiments are actually shown in panel g (i.e. in SH-SY5Y cells or in motor neurons).

Response: Again, the PLA quantitation is from 25 different cells and it is for motor neurons. This has now been mentioned in the figure description.

We again thank the reviewer for critical suggestions, which has significantly helped the readability of the manuscript.

Reviewer #3:

Comment: I thank the authors in question for their thorough and thoughtful rebuttal of my critique of "Mutant FUS causes DNA ligation defects to inhibit repair of oxidative genome damage in Amyotrophic Lateral Sclerosis". I believe that the authors have done an outstanding job to address all concerns and questions. The additional data, experimental controls and discussion points have now generated a compelling study describing the critical role that FUS plays in repairing oxidative breaks and is correlative with patient samples and pathology. Furthermore, the PAR/PARP_i data serves to increase the mechanistic insight of the authors' findings. I am elated to see the connection between mitochondrial DNA integrity and FUS loss; a finding that either is complementary to the present study or may in fact be more important to ALS pathology. While the authors have provided this tidbit of data, a great deal of additional work would need to be performed to extensively elucidate the mechanism of mtDNA degeneration in FUS mutants/KO and its role in the pathology of FUS-ALS. In order to not diminish the impact of this future study, I will treat the mtDNA finding as a "reviewer figure" and for the benefit of the authors, to NOT publish this additional piece data so that it may be included in their future study.

Response: We immensely thank the reviewer for appreciating our efforts in revising this manuscript with new data on the role of PARP-1. We also appreciate the Reviewer for concurring with us about pursuing the mitochondrial implication of FUS toxicity in a separate study.